# ArtVIP: Articulated Digital Assets of Visual Realism, Modular Interaction, and Physical Fidelity for Robot Learning

**Zhao Jin**[1][*]   **Zhengping Che**[1][*]   **Tao Li**[1]   **Zhen Zhao**[1]   **Kun Wu**[1]
**Yuheng Zhang**[1]   **Yinuo Zhao**[1]   **Zehui Liu**[1]   **Qiang Zhang** [1]   **Xiaozhu Ju**[1]
**Jing Tian**[2]   **Yousong Xue**[2]   **Jian Tang**[1][†]
[1]Beijing Innovation Center of Humanoid Robotics
[2]Beijing Institute of Architectural Design
{mustafa.jin, z.che, jian.tang}@x-humanoid.com

## Abstract

Robot learning increasingly relies on simulation to advance complex abilities such as dexterous manipulation and precise interaction, necessitating high-quality digital assets to bridge the sim-to-real gap. However, existing open-source articulated-object datasets for simulation are limited by insufficient visual realism and low physical fidelity, which hinder their utility for training models to master robotic tasks in the real world. To address these challenges, we introduce ArtVIP, a comprehensive open-source dataset comprising high-quality digital-twin articulated objects, accompanied by indoor-scene assets. Crafted by professional 3D modelers adhering to unified standards, ArtVIP ensures visual realism through precise geometric meshes and high-resolution textures, while physical fidelity is achieved via fine-tuned dynamic parameters. Meanwhile, the dataset pioneers embedded modular interaction behaviors within assets and pixel-level affordance annotations. Feature-map visualization and optical motion capture are employed to quantitatively demonstrate ArtVIP's visual and physical fidelity, with its applicability validated across imitation learning and reinforcement learning experiments. Provided in USD format with detailed production guidelines, ArtVIP is fully open-source, benefiting the research community and advancing robot learning research. Our data are available at: `https://huggingface.co/datasets/x-humanoid-robomind/ArtVIP`.

## 1 Introduction

Embodied AI is catalyzing the transformation of robotic systems from constrained laboratory settings (Billard & Kragic, 2019; Spong et al., 2020) to complex, unstructured real-world environments (Brohan et al., 2023b; Zhao et al., 2023; Brohan et al., 2023a). The emergence of large-scale pretrained models (Zhang & Yan, 2023; Kim et al., 2024; Wang et al., 2024b) and novel learning paradigms (Team, 2025; Intelligence, 2025) has ushered in a data-centric era. In this new era, the availability of high-quality data is a critical bottleneck for developing scalable and generalizable embodied intelligence.

---

[*]Equal contribution. [†]Corresponding author.

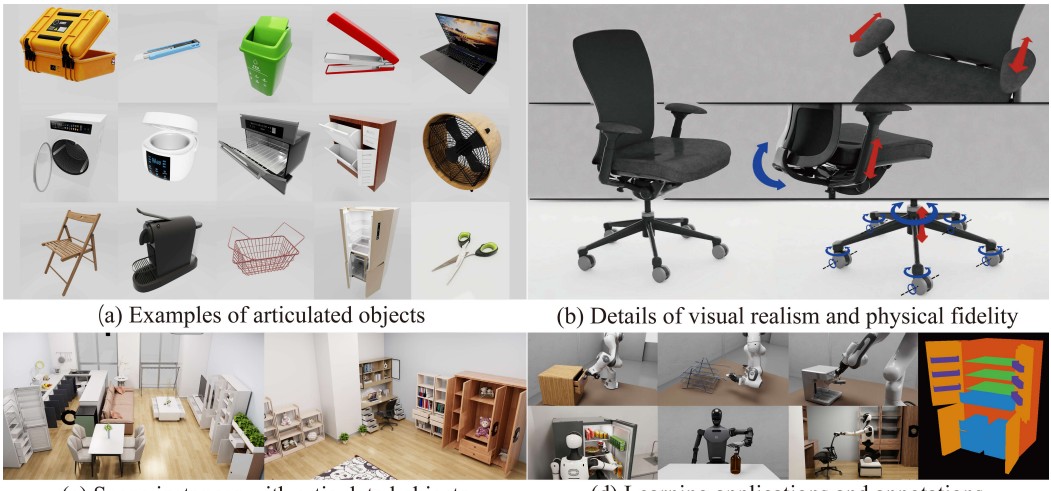

(a) Examples of articulated objects      (b) Details of visual realism and physical fidelity

(c) Scene instances with articulated objects      (d) Learning applications and annotations

Figure 1: ArtVIP spans 9 categories, 37 subcategories, and 992 digital-twin articulated objects. (a) Representative assets across categories and articulation types. (b) High-fidelity physics enables realistic interactions; for example, when pushing an ergonomic chair, its casters rotate accordingly. (c) Six sim-ready scenes in which all objects support real-world–consistent interactions. (d) Pixel-level annotations and sim2real evaluations.

While collecting data and deploying robots in the real world are resource-intensive and challenging to scale, simulation provides an efficient alternative for improving robot learning. Simulation supports imitation learning by collecting unlimited and low-cost training data (Wu et al., 2024) and reinforcement learning by providing virtual environments (Makoviychuk et al., 2021; Torne et al., 2024). Meanwhile, simulations enable rapid deployment and standardized testing (Ramasubramanian et al., 2022; Do et al., 2025) of algorithms without concerns about hardware damage or safety issues. Overall, simulation facilitates the exploration of innovative strategies for robot learning.

High-quality digital assets are vital to simulation for robot learning. Simulation platforms (Koenig & Howard, 2004; Todorov et al., 2012; Kolve et al., 2017; Makoviychuk et al., 2021; Puig et al., 2023) depend on digital assets to accurately represent the real world digitally and to simulate its physical characteristics (Choi et al., 2021). High-quality digital assets can effectively reduce the sim-to-real gap, thereby enhancing the performance of robot learning algorithms. For instance, digital-twin assets, which are virtual replicas created via reverse-modeling techniques, can benefit pre-deployment validation and optimization of robotic systems (Straub et al., 2019; Ramakrishnan et al., 2021). Moreover, high-quality digital assets can serve as training data or seed models for synthetic-asset methods such as 3D reconstruction (Liu et al., 2023a; Li et al., 2020; Sun et al., 2023; Liu et al., 2023b) and domain-randomization (Dai et al., 2024; Ge et al., 2024; Torne et al., 2024) techniques, enhancing the data distribution and providing limitless diversity of objects and environments.

As robot learning turns from mastering simple tasks such as pick-and-grasp to dexterous manipulation and interaction tasks, high-quality articulated-object assets are in great demand. Current open-source articulated-object datasets fail to meet the needs of robot learning. For instance, PartNet-Mobility (Xiang et al., 2020) suffers from limited visual realism and insufficient physical fidelity of dynamic joints. BEHAVIOR-1K (Li et al., 2024a) offers better visual fidelity, but it is locked into the OmniGibson simulator (Li et al., 2024a) and its physical parameters have not been fine-tuned. Moreover, both datasets are largely sourced from internet-searchable 3D model repositories (Inc., 2024a;b) without adhering to consistent modeling standards, leading to inconsistency in quality. Apart from using existing datasets, researchers attempt to obtain simulation assets in other ways, facing further challenges. Retrieval-based methods (Liu et al., 2024b;a) and reconstruction techniques (Chen et al., 2024; Eppner et al., 2024) often inherit stylistic biases from their training data and have limited geometric generalization. More recent pipelines (Qiu et al., 2025; Mandi et al., 2024; Le et al., 2025) introduce promising directions, yet face challenges such as mesh quality variance, segmentation noise, and lack of robust joint parameter tuning.

The main bottleneck for articulated object datasets lies in asset quality rather than quantity; to this end, we identify four key aspects that require careful consideration.

- **Visual Realism.** Assets should be constructed with precise geometric meshes and high-resolution textures to ensure a photorealistic appearance. The amount of triangular faces should be optimized to guarantee real-time simulation performance.
- **Modular Interaction.** Assets should support interactivity (e.g., toggling a switch to turn on a light). These interactions should be modular to enable reuse across scenarios.
- **Physical Fidelity.** Accurate collision geometry and joint dynamics (stiffness, damping, friction) of articulated assets are essential for simulated motion to faithfully reproduce real-world kinematics and dynamics.
- **Simulation Friendliness.** Information that expands simulation usage, such as pixel-level affordance annotations and accompanying scenes, is encouraged. Meanwhile, open-source assets compatible with various simulation platforms and a replicable asset creation process should be provided.

To meet the mentioned requirements, we introduce ArtVIP, a high-quality and readily deployable suite of *__Art__iculated-object digital assets with __V__isual realism, modular __I__nteraction*, and *__P__hysical fidelity*, designed to facilitate the learning and evaluation of diverse manipulation skills such as rotating, clicking, pulling, and pressing. As illustrated in Fig. 1, ArtVIP encompasses both articulated object models and complementary indoor-scene assets, all meticulously authored by professional 3D modelers under a unified asset specification to ensure consistent visual quality and realism. Physical properties are precisely tuned to reproduce real-world dynamics, thereby enhancing the physical fidelity. Furthermore, ArtVIP provides pixel-level affordance annotations and uniquely embeds interaction semantics directly into the assets, enabling modular reuse and scalable behavior modeling.

In conclusion, ArtVIP offers the following contributions:

- We release a collection comprising 9 categories, 37 subcategories, and 992 high-quality digital-twin articulated objects. All assets exhibit both visual realism and physical fidelity, supported by quantitative evaluations.
- We provide digital-twin scene assets and configured scenarios integrating articulated objects within scene for immediate use. Extensive experiments on imitation learning, reinforcement learning, and 3D construction algorithms demonstrate the broader applicability of the assets.
- All assets are provided in the modern USD format and remain compatible with established robotics workflows via conversion to legacy formats such as URDF or MJCF. The detailed production process offers comprehensive guidance to facilitate community adoption and replication.

## 2 RELATED WORKS

**Simulation Platforms.** A typical simulation platform integrates a physics engine (Smith et al., 2005; Todorov et al., 2012; Coumans & Bai, 2016; Corporation, 2025; Tasora et al., 2016) and a rendering engine (Matl, 2019; Chociej et al., 2019; Rojtberg, Pavel and Rogers, David and Streeting, Steve and others, 2001 – 2024). Game engines (Technologies, 2025.05.14; Games, 2025) offer similar features but do not natively support ROS (Quigley et al., 2009; Macenski et al., 2022) for robotics. MuJoCo (Todorov et al., 2012) and Webots (Webots, 2018) excel in simulating rigid body and multi-joint dynamics but prioritize computational efficiency over high-fidelity rendering. Gazebo (Koenig & Howard, 2004), despite its large community and robust integration with ROS, provides outdated rendering performance and exhibits lower accuracy in physical simulation. Frameworks like AI2THOR (Kolve et al., 2017), Habitat (Savva et al., 2019; Szot et al., 2021; Puig et al., 2023) and ALFRED (Shridhar et al., 2020) are designed for mobile manipulation and instruction-following and fail to deliver precise physical interactions. In contrast, Isaac Sim (Nvidia, 2025.05.14) offers the highest-fidelity visual rendering and leverages powerful GPU-parallel physics computation, making it well-suited for robot learning. Other platforms, such as RoboCasa (Nasiriany et al., 2024) (built upon MuJoCo) and OmniGibson (Li et al., 2024a) (built upon Isaac Sim), have become challenging to maintain. Consequently, we developed ArtVIP specifically for Isaac Sim to capitalize on its superior rendering and physics capabilities.

**Datasets for Robot Simulation.** Many datasets provide digital assets suitable for robot simulation. Indoor-scene assets (Straub et al., 2019; Shen et al., 2021; Ramakrishnan et al., 2021; Li et al.,

2022) contribute significantly to robot navigation tasks but lack support for graphical user interface (GUI)-based editing. Object digital asset datasets include ShapeNet (Chang et al., 2015), Objaverse (Deitke et al., 2023) and other digital-twin datasets (Kuang et al., 2023; Dong et al., 2025). However, these assets can only function as rigid bodies in simulations, preventing robots from performing articulated manipulation tasks with them. Limited studies have addressed articulated object assets. PartNet-Mobility (Xiang et al., 2020) provides 2,346 articulated-object assets across 46 categories, with many assets suffering from unsmoothed geometric surfaces, low rendering quality, and imprecise dynamic joints. RoboCasa (Nasiriany et al., 2024) offers 2,508 digital assets, but only 24 are articulated objects. BEHAVIOR-1K (Li et al., 2024a) includes 543 articulated-object assets with improved visual fidelity, yet all assets are encrypted and accessible only through OmniGibson. These limitations underscore the need for a high-quality, open-source articulated-object dataset.

**Articulated Objects Construction and Generation Methods.** Construction methods (Liu et al., 2024a; Chen et al., 2024; Su et al., 2024; Xue et al., 2021; Wang et al., 2024a) can generate articulated objects from images and reduce labor costs. However, these methods perform reliably only on objects with simple joints, such as cabinets and desks, and produce assets with compromised visual realism. Generative methods (Yang et al., 2022; Liu et al., 2023c; Long et al., 2023; Xu et al., 2023; Koo et al., 2024) are currently limited to static rigid-body objects. These assets often exhibit distorted and implausible meshes, coupled with poor rendering quality. The absence of support for articulated objects in generative methods further limits their applicability to robot learning tasks.

## 3 ArtVIP Collection and Methodology

Existing datasets are largely sourced from pre-made models from public repositories. This leads to inconsistent modeling quality, disorganized part hierarchies, and non-standardized coordinate systems, all of which typically require manual preprocessing for simulation use. While current generative and reconstruction methods can easily scale up, they are still not mature enough to ensure quality. Given these constraints, we opted to prioritize fidelity over scale at this stage. ArtVIP emphasizes both visual realism and physical fidelity across a comprehensive collection of articulated objects. It covers 9 categories and 37 subcategories, encompassing 992 articulated assets (see Appendix Sec. A). Complementary sim-ready scenes (see Appendix Sec. B) and pixel-level annotations (see Appendix Sec. C) are also provided.

### 3.1 Visual Realism

To ensure visual realism, professional 3D modelers follow unified modeling and assembly guidelines when manually crafting articulated objects. As shown in Fig. 2, we adopt a top-down mechanical modeling approach that decomposes each articulated object into three hierarchical levels: assembly, module, and mesh. An assembly constitutes the complete functional unit, encompassing multiple

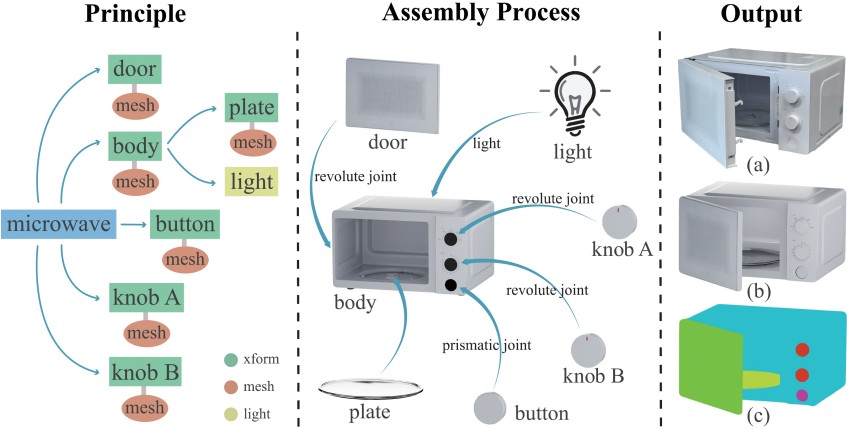

Figure 2: An asset in ArtVIP. **Left:** Top-down assembly principle. **Middle:** Assembly process. **Right:** Comparison between the real object (a) with its digital-twin (b), and annotations (c).

modules and meshes. Modelers first establish the assembly's base coordinate frame at the geometric center of the object's bottom surface. Guided by the assembly's affordances, functionality, and joint locations, they partition it into rigid-body modules of the Xform type, which expose dynamic information such as transforms, velocities, and world coordinates. Each rigid-body module contains mesh parts that provide geometric detail, visual appearance, and static physical properties, including collision shapes and mass. Modelers follow strict rules regarding meshes, textures, and materials (see Appendix Sec. D) to ensure visual realism. After modeling individual meshes, they assemble them bottom-up—mesh, module, assembly—and integrate dynamic motion by connecting modules with joints (middle panel of Fig. 2), ensuring the asset preserves intended affordances and appearance. Finally, for the finished asset (right panel of Fig. 2), each module is annotated with pixel-level labels to enable precise identification of interaction affordances.

## 3.2 Physical Fidelity

In addition to visual realism, physical fidelity plays a critical role in reducing the sim-to-real gap. Optimized collision modeling ensures accurate rigid-body contact, improving precision in tasks such as grasping handles and other force-mediated contact scenarios. Similarly, joint optimization yields precise joint dynamics, improving the fidelity of articulated components' motion trajectories during fine-grained operations (e.g., opening cabinet doors or pressing switches). ArtVIP adopts the following processes.

**Collision.** To strike a balance between physical fidelity, interaction consistency, and computational efficiency, ArtVIP represents each mesh's collision shape using a mix of convex hulls, convex decomposition, and fine-tuned collision meshes. For relatively regular or simple geometry, ArtVIP relies on Isaac Sim's default convex hull generation. When a complex mesh can be decomposed without sacrificing its affordance, 3D modelers split its collision volume into multiple primitive meshes (e.g., cubes, cylinders). If neither a convex hull nor fine-tuned collision suffices, ArtVIP employs Isaac Sim's built-in convex decomposition tool, which leverages mesh normals and related methods to produce accurate collision geometry.

**Joints.** To achieve physical fidelity of dynamic joints and simulate variable joint motions in the real world, we enhance the joint drive equation (NVIDIA, 2025) originally provided by Isaac Sim:

$$\tau = K(q) \cdot (q - q_{\text{target}}(q)) + D \cdot (\dot{q} - \dot{q}_{\text{target}}(q)) \tag{1}$$

where $\tau$ denotes the generalized force or torque applied to drive the joint; $q$ and $\dot{q}$ are the joint position and velocity, respectively; $D$ denotes damping; and $K$ denotes stiffness. While this equation models basic joint motions, it does not fully capture complex joint dynamics observed in the real world. For complex joints such as door closers and light switches, $\tau$ may vary with $q$ and $\dot{q}$. To accommodate these cases, we parameterize $K$ and the target terms as functions of $q$, and allow dependence on $\dot{q}$ when needed. The details are described in the Appendix Sec. E.

## 3.3 Modular Interaction

A key innovation of this work is embedding customizable behaviors directly within each asset to enable interactive functionality without writing additional code.

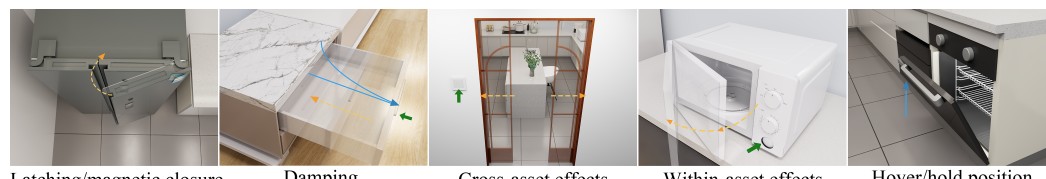

Latching/magnetic closure     Damping     Cross-asset effects     Within-asset effects     Hover/hold position

Figure 3: Green arrows denote applied force, yellow dashed lines indicate object motion, and blue arrows show damping. From left to right: (i) latching/magnetic closure — the door automatically closes when near shut; (ii) damping — the damping magnitude increases as the drawer is pushed in; (iii) cross-asset effects — triggering the switch opens the door; (iv) within-asset effects — pressing the microwave button opens the door and turns on the interior light; (v) hover/hold position — the oven door can hold at any angle.

**Reproducing complex interactions.** We abstract five canonical behavior primitives (Fig. 3) for articulated objects and instantiate them across ArtVIP, covering 394 assets and more than 900 joints.

- *Latching/magnetic closure:* Simulates automatic self-closing when the articulation enters a capture angle range, driven by magnetic attraction or mechanical spring/closer assemblies; once captured, a closing torque is applied until fully latched. Examples include refrigerator doors (self-closing hinge with magnetic gasket) and doors equipped with overhead closers.
- *Damping:* Simulates sliding components and rotational hinges whose effective damping peaks near the closed position and varies smoothly along the motion, enabling gentle starts and stops. Examples include nightstand drawers, dishwashers, and cabinets.
- *Cross-asset effects:* Simulates trigger-based coupling between distinct objects, allowing one object's state or event to drive another's behavior. Examples include button-triggered door opening and light switching.
- *Within-asset effects:* Simulates instantaneous, mechanism-internal triggers. For example, pressing a microwave button pops the door open; similar behaviors occur in foot-pedal trash bins and height-adjustable desks.
- *Hover/hold position:* Simulates static-friction-mediated holding in sliding or rotational joints so that, once external forces are removed, the mechanism can remain at any intermediate pose. Examples include oven doors and drawers.

**Improving asset reusability.** Enhancing simulation development efficiency hinges on modularizing digital assets and maximizing their reusability. Our approach binds behaviors to assets at design time: researchers and artists can simply import the USD file and instantly obtain interaction affordances. This modular, reusable design reduces development overhead and accelerates algorithm iteration, allowing researchers to focus on advancing embodied AI rather than asset programming.

## 4    EVALUATION

We evaluate ArtVIP along two axes: visual realism and physical fidelity, using quantitative comparisons in simulation and the real world. Comparison with generated assets are detailed in Appendix Sec. L.

### 4.1    VISUAL REALISM EVALUATIONS

We conduct a comparative analysis of ArtVIP, BEHAVIOR-1K, and PartNet-Mobility (see Appendix Sec. F for the detailed chart). As shown on the right of Fig. 4, both BEHAVIOR-1K and PartNet-Mobility exhibit distorted geometry and implausible appearance. In addition, we quantify geometric detail via triangle count, evaluate reconstruction performance, and visualize feature distributions to assess visual realism.

**Geometric Detail.** Meshes built from densely triangular faces preserve the core geometric detail. A high count of triangular faces improves surface smoothness and minimizes faceting. The left of Fig. 4 illustrates the comparison results on object categories that appear in all three datasets, demonstrating the rich geometric detail in ArtVIP. More analysis and relative profiling are in the Appendix Sec. G.

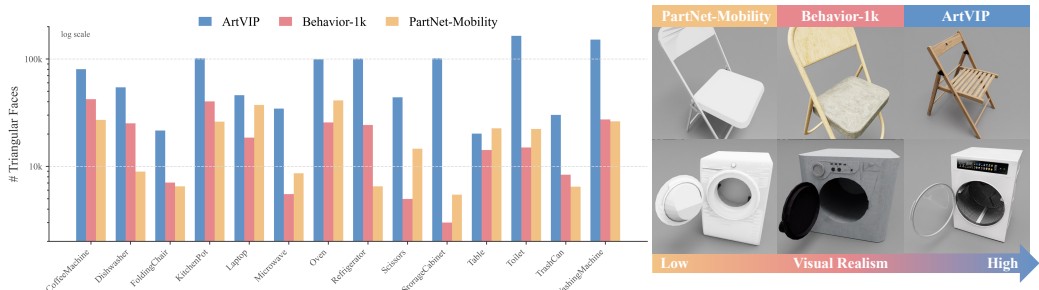

Figure 4: **Left:** Comparison of triangle count. **Right:** Rendering comparison.

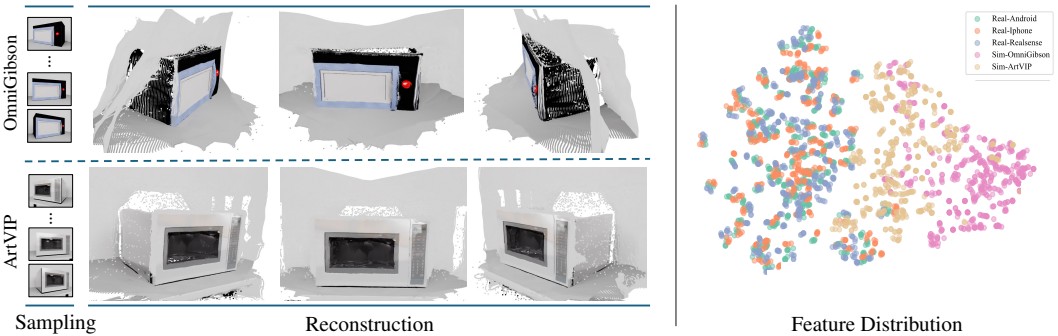

Figure 5: **Left:** Reconstruction of a microwave. OmniGibson yields poor results due to weak visual appearance, while ArtVIP enables better reconstruction via more realistic details. **Right:** CLIP-based (Radford et al., 2021) feature distribution. Each color denotes a data source and ArtVIP features align more closely with real-world data.

**Reconstruction Performance Evaluation.** To assess differences in reconstruction quality across data assets, we conducted experiments using VGGT (Wang et al., 2025), a widely adopted method that has demonstrated strong generalization in real-world reconstruction tasks. Using identical multi-view sampling strategies on the OmniGibson and ArtVIP assets, we generated reconstruction inputs, with results shown on the left portion of Fig. 5. Reconstructions from ArtVIP assets exhibit higher structural fidelity and finer detail preservation compared to those from OmniGibson. This suggests that ArtVIP's more realistic geometry and material representation enhance the quality and compatibility of sampled images for reconstruction tasks. The results underscore the role of high-fidelity assets in supporting viewpoint diversity and accurate structure recovery.

**Feature Distribution Visualization Analysis.** To verify the visual realism of ArtVIP assets, we randomly sampled 100 3D models and selected corresponding or semantically similar objects from OmniGibson and the real world for comparison. Real-world images were captured using three devices (an Android phone, an iPhone, and an Intel RealSense D435) under multi-view settings. In Isaac Sim, we rendered samples of the ArtVIP and OmniGibson assets using matched camera viewpoints to ensure consistency across domains. We applied t-SNE (Van der Maaten & Hinton, 2008) to visualize the extracted CLIP (Radford et al., 2021) features. As shown on the right portion of Fig. 5, ArtVIP features align more closely with real-world data, indicating higher consistency in visual semantics, texture, and material. This fidelity enhances the value of ArtVIP for simulation-to-reality transfer in downstream tasks.

## 4.2 PHYSICAL FIDELITY AND INTERACTION EVALUATIONS

To demonstrate the physical fidelity of joint motion within articulated objects, we employed an optical tracking system (0.1 mm spatial resolution and 90 Hz sampling rate) to record motion trajectories of joints on real-world objects. These recordings were compared with the joint motions of their corresponding digital-twin articulated objects in simulation to evaluate the discrepancy between simulated and real-world joint behavior. We test in a common scenario where joint motion triggered by external force. More setting descriptions and evaluation results are described in the Appendix Sec. H.

As shown in Fig. 6, in the real-world experiment, horizontal pulling forces of 1 N, 1.5 N, 2 N, and 2.5 N were applied to the drawer by suspending calibrated weights from the end of the fixed pulley system, ensuring consistent force direction. The drawer's displacement in the XY plane was recorded in real time. In the simulation environment, two configurations were evaluated: one with default joint parameters and the other with optimized parameters. Both were subjected to the same force configuration as the real-world setup, and the spatial trajectories of the drawer's keypoints were tracked. The close agreement between the displacement obtained from simulation and real-world experiments, as shown in the right of Fig. 6, demonstrates the physical fidelity of the joints in ArtVIP.

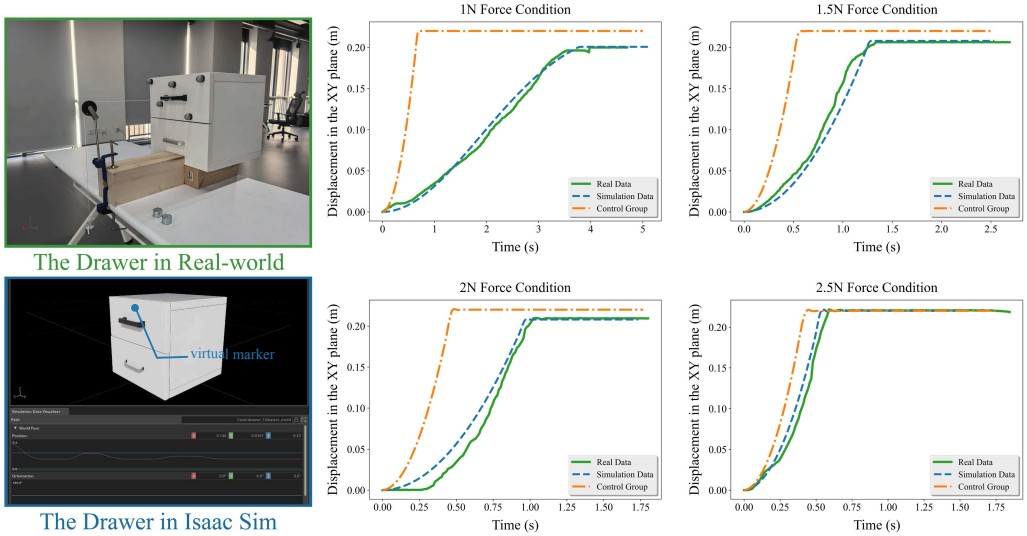

Figure 6: **Left:** Digital-twin asset examples in real-world and simulation. **Right:** Analysis of the drawer's displacement driven by different forces.

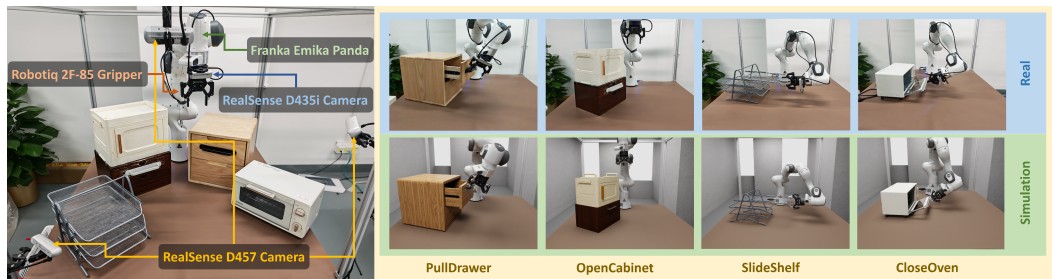

Figure 7: Experimental Setup. We conducted 4 real-world tasks for imitation learning.

# 5 APPLICATIONS

To further verify the capability of ArtVIP in supporting downstream robotic learning tasks, we conducted extensive experiments in both the real-world and simulated environments following two primary paradigms in robotic learning: Imitation Learning and Reinforcement Learning.

## 5.1 IMITATION LEARNING IN REAL WORLD ENVIRONMENTS

**Experimental Setup.** As illustrated in Fig. 7, we used a Franka robotic arm equipped with a Robotiq 2F-85 gripper and four RealSense cameras to create the real-world experimental environment. These cameras include three external RealSense D457 cameras (placed on the left, right, and top of the table) and one hand-eye RealSense D435i camera mounted at the wrist of the robotic arm. For simulation, we used Isaac Sim and replicated this real-world setup, including the Franka robotic arm, the operating table, camera settings, and the manipulated objects from ArtVIP. We constructed the simulated scene to match the real-world experiment environment as closely as possible.

**Task Design and Data Collection.** As shown in Fig. 7, we design four challenging articulated-object manipulation tasks: (1) **PullDrawer**, (2) **OpenCabinet**, (3) **SlideShelf**, and (4) **CloseOven**. These tasks demand precise and flexible motions, including rotation, angled pushing, and horizontal translation (see Appendix Sec. I). Data was collected via teleoperation in both real and simulated environments, where articulated objects were randomly placed within a predefined workspace and human operators completed each task. For each task, we gathered 100 successful trajectories in

Table 1: Success rates of ACT and DP across dataset settings: RO (real-only), SO (sim-only), and RSM variants for all tasks.

| Method | Dataset | PullDrawer | OpenCabinet | SlideShelf | CloseOven |
|---|---|---|---|---|---|
| ACT (Zhao et al., 2023) | RO | 64% | 34% | 27% | 58% |
| | SO | 39% | 12% | 13% | 23% |
| | RSM100+10 | 64% | 36% | 26% | 59% |
| | RSM100+20 | 68% | 38% | 27% | 60% |
| | RSM100+50 | 78% | 44% | 32% | 66% |
| | RSM100+100 | 81% | 46% | 36% | 68% |
| DP (Chi et al., 2023) | RO | 66% | 49% | 44% | 66% |
| | SO | 20% | 10% | 18% | 28% |
| | RSM100+10 | 65% | 53% | 47% | 67% |
| | RSM100+20 | 69% | 58% | 53% | 70% |
| | RSM100+50 | 73% | 62% | 56% | 73% |
| | RSM100+100 | 79% | 66% | 59% | 78% |

the real world and 100 in simulation. Each trajectory includes RGB streams from four camera viewpoints and full proprioceptive robot states (e.g., joint positions) throughout execution.

**Imitation Learning Algorithm.** We used two canonical imitation learning baselines, Action Chunking Transformer (ACT) (Zhao et al., 2023) and Diffusion Policy (DP) (Chi et al., 2023), to train the robotic policies for the articulated object manipulation task (more details in Appendix Sec. I).

**Experimental Results on Imitation Learning.** For each of the four articulated-object manipulation tasks, we trained ACT and DP under the following dataset settings: **(1) Real-Only (RO)**: 100 real-world trajectories; **(2) Sim-Only (SO)**: 100 simulated trajectories; **(3) Real–Sim–Mixed (RSM100+10/20/50/100)**: 100 real-world + 10, 20, 50, 100 simulated trajectories. For each experiment, we trained ACT and DP for 50k gradient descent iterations with three different random seeds, and evaluated the final checkpoint from each run with 60 rollouts to compute per-task success rates.

Tab. 1 summarizes success rates for ACT and DP under three dataset settings (RO, SO, RSM). We highlight three findings: (1) **Simulation-trained models achieve zero-shot success in the real world** (e.g., ACT 39% on PullDrawer), reflecting ArtVIP's high-fidelity visuals and physics that reduce the sim-to-real gap. (2) With equal data volume, **real-world training outperforms simulation** (e.g., DP 49% vs. 10% on OpenCabinet), underscoring persistent sim-to-real challenges. (3) **Mixing real and simulated data boosts performance** (e.g., SlideShelf: DP from 44% to 59%), indicating that articulated assets in ArtVIP align well with real-world data distributions.

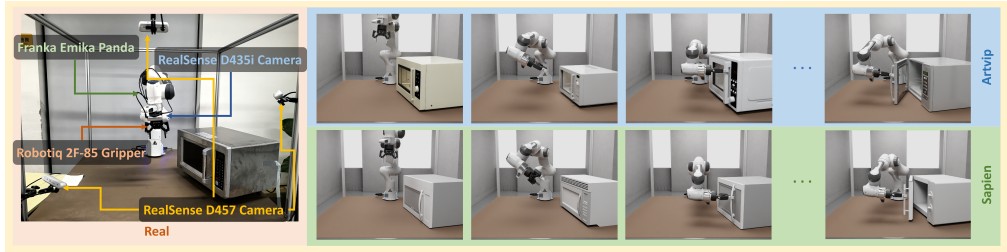

Figure 8: We collect data on five microwaves each from ArtVIP and PartNet-Mobility.

## 5.2 COMPARISON WITH OTHER ASSETS VIA IMITATION LEARNING.

To validate the quality of ArtVIP assets against other datasets, we conduct a digital-cousin comparison with PartNet-Mobility (Fig. 8). We select five microwave ovens from PartNet-Mobility and five from ArtVIP. We select microwaves with pull-to-open doors and deliberately exclude button-triggered opening, as it is operationally trivial. We collect data via teleoperation following the same procedure as in the digital-twin experiments mentioned before, obtaining 100 simulated trajectories per microwave (500 in total). For the real-world task, we purchase an unseen microwave oven for

Table 2: Success rates of ACT and DP across dataset settings: RO (real-only), SO (sim-only), and RSM100+500, comparing ArtVIP and PartNet-Mobility on the microwave door-pull task.

| Method | Dataset | ArtVIP | PartNet-Mobility |
|--------|---------|--------|------------------|
| ACT | RO | | 56% |
| | SO | 41% | 32% |
| | RSM100+500 | 79% | 68% |
| DP | RO | | 62% |
| | SO | 45% | 35% |
| | RSM100+500 | 83% | 70% |

which neither PartNet-Mobility nor ArtVIP provides a corresponding digital-twin model. We train ACT and DP under the following dataset settings: **(1) Real-Only (RO)**: 100 real-world trajectories; **(2) Sim-Only (SO)**: 500 simulated trajectories; **(3) Real–Sim–Mixed (RSM100+500)**: 100 real-world + 500 simulated trajectories. All runs use the same training hyperparameters.

Tab. 2 summarizes success rates across three dataset settings. We highlight: **higher-quality ArtVIP assets yield stronger zero-shot sim-to-real transfer under SO and higher success under RSM**, supporting the conclusion that higher-quality assets reduce the sim-to-real gap and lead to higher success rates.

### 5.3 Reinforcement Learning in High-Fidelity Simulators

Reinforcement learning (RL) requires training environments that mirror real-world physical and perceptual complexity. To evaluate ArtVIP, we design a CloseTrashcan task with a Franka arm and train a two-stage agent in Isaac Sim using EAGLE (Zhao et al., 2025) (Appendix Sec. J). We first train a PPO expert (Schulman et al., 2017) with low-level state inputs, then distill it into a visuomotor policy, applying EAGLE's self-supervised attention masks and control-aware augmentation. We additionally use RandomConv (Lee et al., 2019) and Visual Matching (Li et al., 2024b) to reduce the background gap between simulation and the real world.

Table 3: EAGLE vs. vision-based PPO: success rate across training checkpoints (k).

| Method | Training Iterations (k) | | | | |
|--------|-----|-----|-----|-----|-----|
| | 100 | 200 | 300 | 400 | 500 |
| EAGLE | 0.23 | 0.28 | 0.73 | 0.85 | 0.98 |
| Vision-based PPO | 0.16 | 0.19 | 0.21 | 0.22 | 0.24 |

We train in simulation and deploy the same policy in the real world. Tab. 3 reports success rates (more details in Appendix Sec. K) at five checkpoints (300k–500k iterations), evaluated with 100 simulation trials and 30 real-world trials under varied initial object poses. The Pearson correlation coefficient is 0.9886, indicating a strong linear relationship between simulated and real-world performance, and supporting the fidelity of ArtVIP.

## 6 Limitation and Conclusion

We introduced ArtVIP, a high-quality dataset of articulated objects for robotic manipulation, featuring visual realism, accurate physical properties, and modular interaction capabilities. We validated its quality via diverse evaluations and demonstrated effectiveness in both imitation learning and reinforcement learning. Scaling remains bottlenecked by intensive human labor for asset modeling; future work will explore generative methods to automate synthesis, reduce manual effort, and broaden object diversity.

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

## A  ARTICULATED OBJECTS

ArtVIP comprises 992 articulated objects, encompassing 9 categories and 37 subcategories, with a total of 2156 prismatic joints and 1809 revolute joints. The detailed breakdown, including approximate human labor time, is presented in Tab. 4.

## B  SCENES

We provide sim-ready complex, dynamic environments—six in total: childrenroom, diningroom, kitchen, kitchen with parlor, large livingroom, and small livingroom (see Fig. 9 for two example scenes). Every object in these environments, including fixed furniture, supports physical interaction. This includes switches, small appliances, plush toys, laptops, books, spice jars, and more. For example, the kitchen environment contains a total of 65 joints, and all objects can be used just like their real-world counterparts. Robots can operate the light switch on the wall, open the refrigerator door, place items on shelves, or challenge their motion capabilities by crouching to open drawers beneath the stove top. Additionally, users can freely place the 992 articulated objects provided in ArtVIP into any of these environments via the Isaac Sim GUI, enabling the creation of rich robot interaction scenarios such as grasping, pulling, pressing, and placing. Users can also utilize open-source tools like mjcf2usd and urdf2usd to convert assets from other datasets into the USD format, allowing seamless integration with ArtVIP assets. This kind of sim-ready, complex environment is currently unique to ArtVIP. Moreover, the ability to edit and save assets directly through a GUI reflects an open-source spirit that is not yet common in other datasets.

## C  ANNOTATIONS

Annotations in ArtVIP provide objective descriptions of object parts, thereby supporting robots' ability to infer task-appropriate interaction behaviors. We further argue that annotations are most meaningful when aligned with consistent modeling standards. For example, for a desk, modelers often merge the legs and tabletop into a single mesh, which limits part-level annotation based on distinct interaction functions. To address this, we highlight functional components in Tab. 5 that frequently participate in interactions yet are commonly overlooked during mesh segmentation. An example segmentation is shown in Fig. 10.

## D  MODELING STANDARDS

In simulation systems, the use of high-quality meshes, textures, and materials confers several advantages. High-fidelity visuals reduce the disparity between simulation and reality (Nesti et al., 2025), thereby narrowing the sim-to-real gap and enabling robotic policies to be deployed in real-world environments with minimal or even zero-shot adaptation (Han et al., 2025; Embley-Riches et al., 2025). Photorealistic simulation data can be employed to train and validate visual perception algorithms, such as object detection, semantic segmentation, and SLAM. Moreover, realistic models not

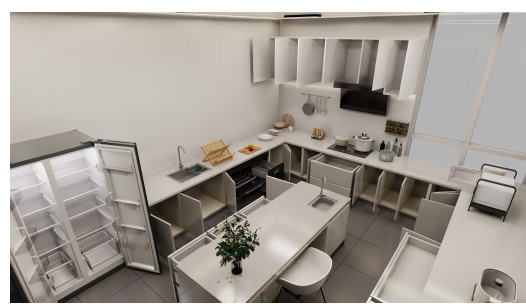
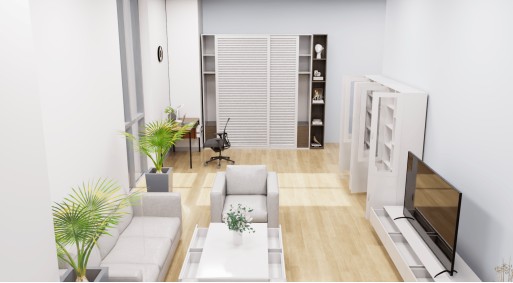

|  (a) Kitchen | (b) Small livingroom |

Figure 9: Scenes: all articulated joints in the open state.

Table 4: Detailed breakdown of object categories, modeling time (each), physics tuning time (each), and count.

| category | subcategories | modeling time | physics tuning time | count |
|---|---|---|---|---|
| furniture | chair | 2h | 0.3h | 23 |
| | table | 1.5h | 0.2h | 131 |
| | cabinet | 3.1h | 0.4h | 183 |
| | cupboard | 15h | 2h | 11 |
| | bed | 3h | 0.3h | 28 |
| | home decor | 2.5h | 0.3h | 30 |
| kitchenware | cookware | 1.9h | 0.3h | 81 |
| kitchen appliances | coffee machine | 3h | 0.3h | 14 |
| | built-in oven | 5h | 0.3h | 14 |
| | microwave | 3h | 0.3h | 8 |
| | oven | 4h | 0.3h | 11 |
| | dishwasher | 5h | 0.4h | 19 |
| | water dispenser | 3h | 0.3h | 6 |
| | rice cooker | 3h | 0.3h | 14 |
| | fridge | 6h | 0.5h | 22 |
| | juicer | 4h | 0.3h | 6 |
| fixtures | faucet | 2h | 0.2h | 14 |
| | toilet | 4h | 0.4h | 14 |
| | door | 2h | 0.3h | 10 |
| appliances | computer | 2.5h | 0.3h | 13 |
| | fan | 1.8h | 0.3h | 34 |
| | air conditioner | 4h | 0.3h | 3 |
| | washing machine | 5.7h | 0.5h | 30 |
| | speaker | 1.5h | 0.3h | 14 |
| | floor lamp | 1h | 0.3h | 28 |
| cleaning tools | mop | 2h | 0.3h | 8 |
| | pump bottle | 2h | 0.3h | 14 |
| | trash can | 2h | 0.3h | 18 |
| stationery | scissors | 1h | 0.2h | 28 |
| | stapler | 1.5h | 0.2h | 11 |
| | utility knife | 1h | 0.2h | 19 |
| | folder | 0.5h | 0.2h | 8 |
| storage | storage box | 2h | 0.3h | 25 |
| | toolbox | 2.5h | 0.3h | 22 |
| | cardboard box | 1.5h | 0.2h | 28 |
| Mechanical equipment | electrical equipment | 3h | 0.3h | 17 |
| | non-electrical equipment | 3h | 0.3h | 33 |

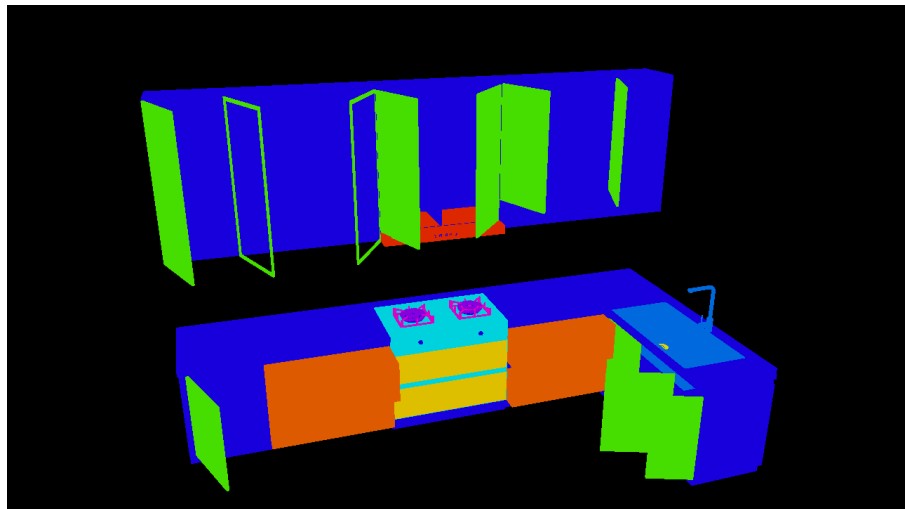

Figure 10: Segmentation result of the kitchen scene.

only enhance visual fidelity but also improve interaction effects within simulations. When robots perform actions such as grasping, collision, or force-based interactions, accurate geometry ensures stable and reliable feedback. To achieve photorealistic appearance and minimize the sim-to-real visual gap, we addressed the following standards:

**Mesh.** Manifold meshes form the core geometric foundation of each asset, defining the object's overall contour and spatial occupancy. These meshes are critical for generating collision bodies that maintain accuracy in physical interactions. ArtVIP ensures that mesh details produce smooth surfaces and lifelike contours, avoiding jagged or blocky appearances. Additionally, through normal vector optimization algorithms, redundant vertices are merged, reducing geometric data volume and thereby alleviating computational burdens in simulation.

**Texture.** Textures are mapped onto mesh surfaces via UV coordinates to provide visual details. ArtVIP employs high-resolution textures to capture fine surface characteristics, such as the metallic sheen of a refrigerator or the subtle grain of wood on a chair. Furthermore, textures are meticulously aligned with the UV map to prevent stretching, distortion, or visible seams.

**Material.** A material is a collection of rendering parameters, including references to textures, that defines how an object's surface responds to light. ArtVIP leverages RTX Renderer (Nvidia, 2025b) in Isaac Sim and adopts Physically Based Rendering (PBR) (Nvidia, 2025a) to accurately simulate diffuse and specular reflections, enabling rendering effects such as roughness and emissive properties. This approach allows for the realistic representation of diverse materials, achieving true-to-life visual fidelity.

## E  PHYSICAL FIDELITY OF JOINTS

To achieve physical fidelity of dynamic joint and simulate variable joints motions in the real world, we enhance the joint drive equation originally provided by Isaac Sim:

$$\tau = K(q) \cdot (q - q_{\text{target}}(q)) + D \cdot (\dot{q} - \dot{q}_{\text{target}}(q)) \tag{2}$$

where $\tau$ represents the force($F$) and torque($T$) applied to drive the joint, $q$ and $\dot{q}$ are the joint position and velocity, respectively, $D$ donates damping, and $K$ donates stiffness. While this equation can model basic joint motions, it fails to fully replicate complex dynamic joint motions in the real world. For complex joints such as door closers and light switches, $\tau$ may vary with $q$ and $\dot{q}$. To accommodate the above situations, we design functions of $q$ and $\dot{q}$.

**Impact from $\dot{q}$.** Friction must be accounted for in simulation and cannot be modeled as a constant. It imposes resistance to the force generated by the joint drive $\tau$, and we propose the following equation

Table 5: Annotation labels and descriptions in ArtVIP.

| Label | Description |
|---|---|
| armrest | Chair armrest |
| backrest | Chair backrest |
| ball_handle | Handle for lifting the main body, such as the handle of a toolbox |
| blade | Blade of a utility knife, scissors, or fan blades |
| body | Parts that need labeling excluding base and lid |
| button | Applies to all push-button switch components of models |
| door | Door of cabinets, refrigerators, ovens, etc. |
| drawer | Drawer of cabinets, refrigerators, toolboxes, etc. |
| front_cover | Cover of a folder |
| fun_guard | Fan protective cover |
| handle | Any handles |
| headrest | Chair headrest |
| jaw | Head of pliers, the part that contacts the gripped item |
| keyboard | Computer keyboard |
| knob | Applies to all rotary switch components of models |
| lid | Such as cardboard box lid, electric steamer lid, trash can lid |
| light | All types of lights |
| mop_head | Mop head |
| pedal | Foot pedal, such as on a step-on trash can |
| pipe | Water pipe part of faucet |
| plate | All types of plates |
| pole | Rod-shaped component |
| portafilter | A handle holds the coffee grounds |
| pot | Inner pot of rice cookers, steamers, etc. |
| rack | Rack in an oven, refrigerator door shelf |
| roller | Washing machine drum |
| screen | Electronic product screen |
| seat | Chair seat |
| shelf | Shelf part of cabinets, refrigerators, etc. |
| spout | Spout of a pump bottle, water dispenser, etc. |
| stapler_magazine | Staple compartment of a stapler |
| tabletop | Top surface of a table |
| toilet_seat | Toilet seat |
| touch_pad | Computer touchpad |
| wheel | Chair wheels |

with three different conditions:

$$
F_{\text{friction}}(\dot{q}) = \begin{cases} -F_{\text{ext}} & \dot{q} = 0 \text{ and } |F_{\text{ext}}| \leq \mu_s \cdot (|F| + |T|) & \text{(3a)} \\ -\mu_s \cdot (|F| + |T|) \cdot \text{sign}(F_{\text{ext}}) & \dot{q} = 0 \text{ and } |F_{\text{ext}}| > \mu_s \cdot (|F| + |T|) & \text{(3b)} \\ -D \cdot \dot{q} \cdot \text{sign}(\dot{q}) & \dot{q} \neq 0 & \text{(3c)} \end{cases}
$$

We illustrate the friction from static friction, to maximum static friction, and finally to dynamic friction, corresponding to conditions from Eqn. equation 3a through Eqn. equation 3c. $F_{\text{ext}}$ denotes the static friction. The coefficient $u_s$ denotes the static friction coefficient, which can be configured in Isaac Sim via the `Joint Friction` parameter. The *sign* function ensures that the frictional force is applied in the correct direction.

**Impact from** $q$**.** The latch release mechanism exemplifies the position-dependent joint drive, we analyze a button-actuated trash bin lid mechanism. When the button is depressed, it triggers a linkage to retract the spring-loaded latch, enabling the lid to freely rotate under torsional spring torque to $q_{\text{upper\_bound}}$.

$$
q_{\text{target}}(q) = \begin{cases} q_{\text{upper\_bound}} & \text{if } q > q_{\text{threshold}} \text{ and } S_{\text{open}} = 1 & \text{(4a)} \\ q_{\text{lower\_bound}} & \text{if } q < q_{\text{threshold}} \text{ and } S_{\text{open}} = 0 & \text{(4b)} \end{cases}
$$

We further investigate joint motion with abrupt stiffness variations, exemplified by refrigerator door closers and magnetic latching mechanisms. To maintain static equilibrium in the stationary state, a high stiffness value $k_{\text{high}}$ is employed. When $S_{\text{open}} = 1$ (door opening phase), the stiffness progressively decreases with increasing $q$. Upon exceeding the critical position $q_{\text{threshold}}$, the stiffness reaches its minimum $k_{\text{low}}$, and the joint target position switches to $q_{\text{upper\_bound}}$. During door closure, as $q$ approaches $q_{\text{threshold}}$ from above, the target position abruptly transitions to $q_{\text{lower\_bound}}$, accompanied by an exponential stiffness surge to rapidly complete closure, emulating commercial door closer dynamics. This behavior is formalized as:

$$
K(q) = \begin{cases}
k_{\text{high}}, & q = q_{\text{lower\_bound}} & \text{(5a)} \\
k_{\text{high}} - \alpha q, & \text{if } q_{\text{lower\_bound}} < q \leq q_{\text{threshold}} \text{ and } S_{\text{open}} = 1 & \text{(5b)} \\
k_{\text{low}} + k_{\text{max}} e^{-\lambda q}, & \text{if } q_{\text{lower\_bound}} < q \leq q_{\text{threshold}} \text{ and } S_{\text{open}} = 0 & \text{(5c)} \\
k_{\text{low}}, & q_{\text{threshold}} < q < q_{\text{upper\_bound}} & \text{(5d)}
\end{cases}
$$

## F    CHART COMPARISON WITH EXISTING DATASETS

We present a detailed comparison of ArtVIP with existing articulated-object datasets in Tab. 6.

Table 6: Detailed comparison with existing articulated-object datasets.

|  | Articulated Assets | Prismatic Joints | Revolute Joints | Visual Realism | Physical Fidelity | Modular Interaction |
|---|---|---|---|---|---|---|
| ArtVIP | 992 | 2156 | 1809 | high | high | 394 |
| BEHAVIOR-1K | 545 | 318 | 819 | medium | low | None |
| PartNet-Mobility | 2347 | 7659 | 4312 | low | low | None |

## G    VISUAL REALISM COMPARISON

We present further comparative analysis in Fig. 11. PartNet-Mobility employs the URDF format, with meshes stored in OBJ format and material information defined in MTL files. Although the OBJ files are manually crafted, they frequently exhibit distorted meshes, significantly compromising visual quality. The MTL material format inherently lacks the capability to model physically accurate light reflection, resulting in a lack of environmental realism across all PartNet-Mobility assets. Our analysis reveals that many materials in PartNet-Mobility rely solely on base color for rendering, and the absence of textures substantially degrades the overall rendering quality. Although BEHAVIOR-1K adopts the USD format, which supports physically based rendering (PBR), it still suffers from issues related to distorted meshes and poor texture quality.

To mitigate issues such as distorted meshes and angular surfaces, we employed a high number of triangular faces to ensure smooth surfaces and enhanced geometric detail. For categories such as toilets and refrigerators, ArtVIP significantly surpasses BEHAVIOR-1K and PartNet-Mobility in the number of triangular faces utilized. However, this approach entails a trade-off, as it reduces the simulation frame rate. To address this, we conducted profiling analysis to optimize the simulation frame rate for each object. In our experiments, we selected the kitchen, which contains the highest number of articulated objects, and the living room, which features the most extensive texture rendering, as testing environments. Each asset from ArtVIP was individually placed within these scenes, ensuring that the overall rendering frame rate consistently exceeds 60 Hz (i7-13700, Nvidia 4090, 64 GB).

To study the effect of triangle count, we report comprehensive statistics in Tab. 7 for ArtVIP, PartNet-Mobility, and BEHAVIOR-1K: the average triangle count, the average number of active joints, the average FPS with a single asset, and the average FPS in the kitchen scene. The kitchen scene is the most complex environment, containing 65 actuated joints. ArtVIP and PartNet-Mobility are evaluated in Isaac Sim 5.1. BEHAVIOR-1K assets are encrypted and accessible only through OmniGibson (Isaac Sim 4.5). We attribute the large FPS fluctuations observed for BEHAVIOR-1K to overhead introduced by the derivative framework. Based on the FPS results for ArtVIP and PartNet-Mobility, we conclude: 1) For a single object, under Isaac Sim's iterative optimizations, triangle

Table 7: Category-wise averages for triangle count, active joints, and FPS across datasets.

| Item Category | Avg. Triangle Count | | | Avg. Active Joints | | | Avg. FPS (Single Item) | | | Avg. FPS (In Kitchen) | | |
|---|---|---|---|---|---|---|---|---|---|---|---|---|
| | ArtVIP | PartNet-Mobility | BEHAVIOR-1K | ArtVIP | PartNet-Mobility | BEHAVIOR-1K | ArtVIP | PartNet-Mobility | BEHAVIOR-1K | ArtVIP | PartNet-Mobility | BEHAVIOR-1K |
| Coffee Machine | 80484.8 | 27104.7 | 42256 | 2.2 | 5.759 | 5.5 | 91.97 | 91.95 | 114.77 | 72.01 | 73.83 | 48.33 |
| Microwave | 34494.6 | 8620.5 | 5521 | 4 | 4.313 | 1.857 | 91.93 | 91.88 | 87.51 | 69.38 | 72.95 | 50.04 |
| Oven | 99048 | 41206.2 | 25638 | 4.5 | 6.133 | 1 | 91.98 | 91.95 | 109.97 | 72.66 | 74.64 | 49.4 |
| Dishwasher | 54427.1 | 8932.6 | 25162.2 | 1.429 | 1.333 | 2.5 | 91.94 | 91.95 | 115.2 | 65.75 | 69.95 | 49.73 |
| Rice Cooker | 101573.3 | 26068.7 | 40245.3 | 3.333 | 1.12 | 1 | 91.96 | 91.96 | 115.58 | 70.97 | 74.91 | 48.29 |
| Laptop | 46053.6 | 37378.7 | 18546.3 | 1 | 1 | 1 | 91.97 | 91.93 | 112 | 74.59 | 74.89 | 46.99 |
| Washing Machine | 151705.4 | 26269.8 | 27380.8 | 2.57 | 7.471 | 1.538 | 91.95 | 91.94 | 107.17 | 70.97 | 70.74 | 48.27 |
| Toilet | 164271.6 | 22276.49 | 15011.11 | 3.6 | 2.319 | 2.611 | 91.95 | 91.95 | 120.58 | 74.94 | 74.93 | 47.18 |
| Refrigerator | 100903.8 | 6517 | 24273.4 | 6.25 | 1.682 | 1.538 | 91.94 | 91.96 | 99.55 | 60.78 | 73.89 | 49.82 |
| Table | 20184.7 | 22607.6 | 14210.6 | 5.28 | 3.158 | 2.633 | 91.96 | 91.96 | 116.37 | 68.13 | 71.68 | 47.92 |
| Folding Chair | 21567.5 | 6519.3 | 7064.6 | 2 | 1.231 | 2 | 91.95 | 91.97 | 125.93 | 74.5 | 74.92 | 51.41 |
| Scissors | 43953 | 14601 | 4972 | 2 | 1.963 | 1 | 91.96 | 91.96 | 129.71 | 72.5 | 74.92 | 52.45 |
| Trash Can | 30139.6 | 6468.33 | 8370.17 | 1.77 | 1.971 | 1 | 91.94 | 91.93 | 121.28 | 71.66 | 74.93 | 52.24 |

counts up to approximately 100k and up to 20 active joints have negligible impact on FPS. 2) In complex scenes, both triangle count and the number of active joints reduce FPS.

# H    PHYSICAL FIDELITY AND INTERACTION EVALUATIONS

**Motion Triggered by Latch Release.** To validate the modular interaction within assets, we compared the triggered joint in both real-world and virtual microwave. We conducted button-press experiments in each environment to initiate the door-opening action and recorded the resulting door motion trajectories. In the real-world tests we tracked a marker on the door using the optical tracking system to capture its spatial motion after the button pressed. In the simulation we set a virtual marker at the same position as the real-world marker on the door, and we triggered the door opening via pressing the button as well (for which the activation configured in modular interaction) and logged the virtual marker's trajectories. We performed ten trials in each environment and computed the average spatial trajectory as Fig. 12 shown.

**Motion Triggered by Joint Position Threshold.** Appliances equipped with door closers typically exhibit a dynamic change in motion once the door reaches a certain angle during closing. After arriving at a certain angle, the door closer causes the door to accelerate and snap shut against the appliance body. To evaluate how well the simulation captures this physical transition, we focus on analyzing the door's linear and angular velocities during the transition from the threshold state to full closure. In both the simulation and real-world experiments, a force of no more than 1.0 N is applied when the door is within the threshold range to trigger the door closer mechanism. We then record the kinematic behavior following the activation of the door closer. In the real-world setup, the optical motion capture system is used to track the spatial displacement of markers on the door. Both the simulation and real-world experiments are repeated ten times, and we compute the average spatial trajectories and changes in velocity along the X-axis for quantitative comparison (Fig. 13).

# I    IMITATION LEARNING APPLICATION

**Task Summary.** As shown in Fig. 14, we design four challenging articulated-object manipulation tasks: (1) **PullDrawer**, (2) **OpenCabinet**, (3) **SlideShelf**, and (4) **CloseOven**. These tasks demand precise and flexible motions, including rotation, angled pushing, and horizontal translation. We define these tasks as follows:

- **PullDrawer**. his task requires the robot to insert the gripper into the handle of the drawer, securely press the handle, and gradually pull the drawer out along a linear trajectory using a smooth and consistent motion.
- **OpenCabinet**. For this task, the robotic arm needs to precisely locate the thin vertical handle of the cabinet door. The gripper has to align vertically, firmly grip the handle, and pull the door outward along a curved path while maintaining a stable trajectory.
- **SlideShelf**. This task involves horizontal manipulation of the shelf. First, the gripper needs to rotate around 90 degrees to align parallel to the shelf's direction. It then grips the base of the shelf and moves horizontally, pulling the shelf out along its guide rails in a stable and controlled manner.
- **CloseOven**. To complete this task, the robotic arm needs to close its gripper to push against the bottom edge of the oven door. The arm then rotates and lifts under the door, applying a curved upward force to close the door.

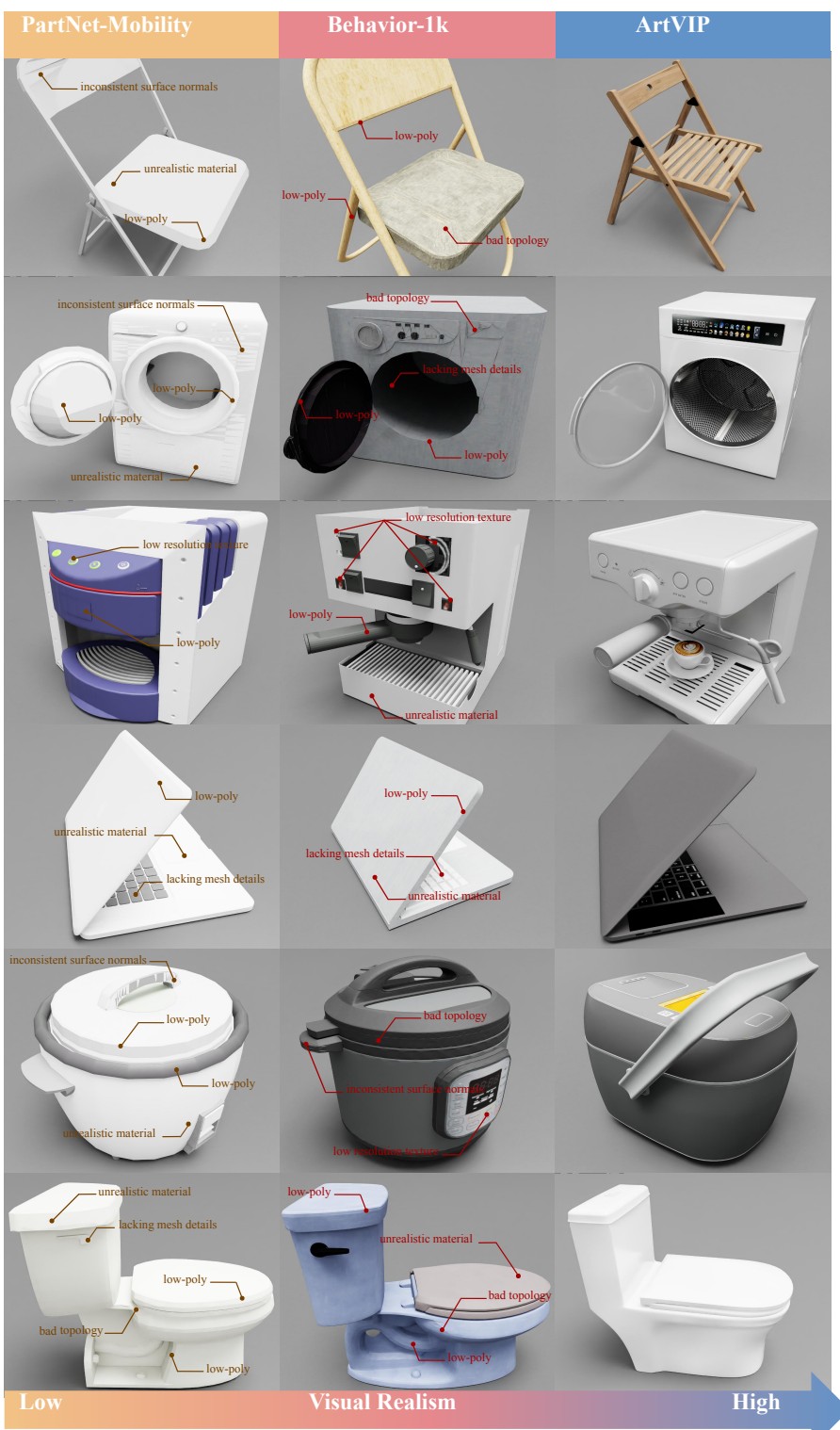

Figure 11: Comparisons of ArtVIP, BEHAVIOR-1K, and PartNet-Mobility.

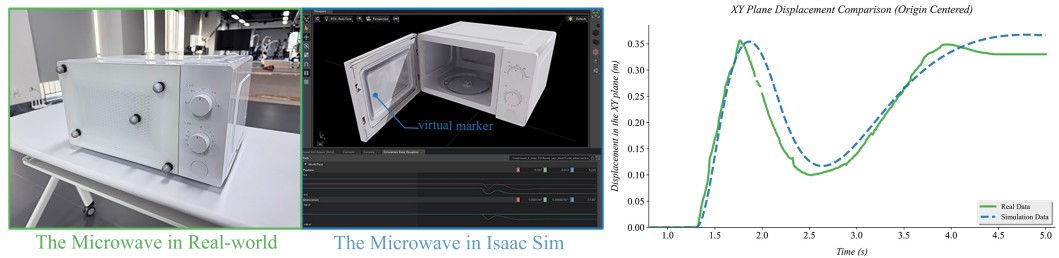

Figure 12: **Left and Middle:** Digital-twin asset examples in real-world and simulation. **Right:** Analysis of the Microwave's displacement.

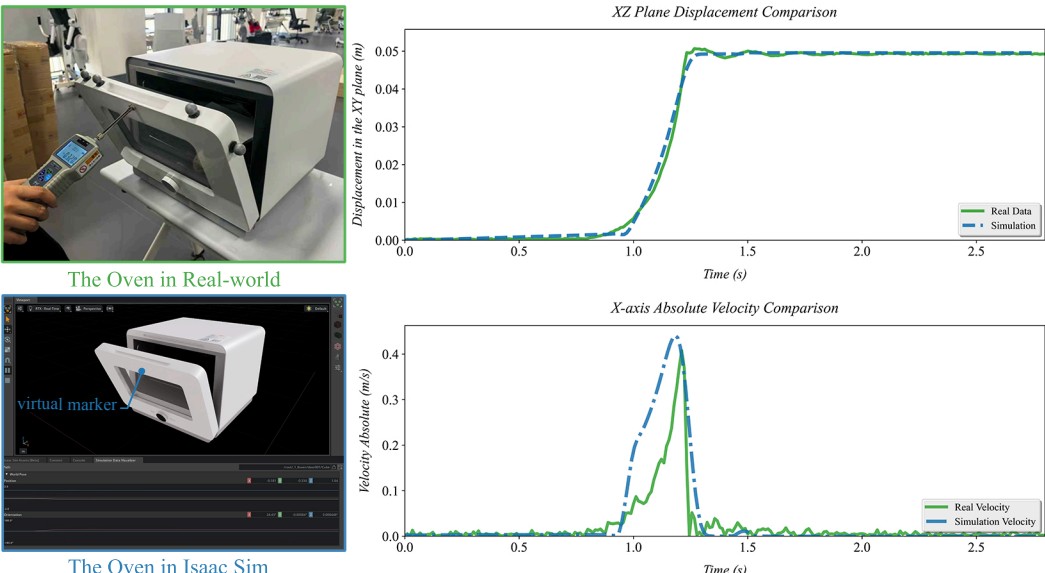

Figure 13: **Left:** Digital-twin asset examples in real-world and simulation. **Right:** Analysis of the oven's displacement.

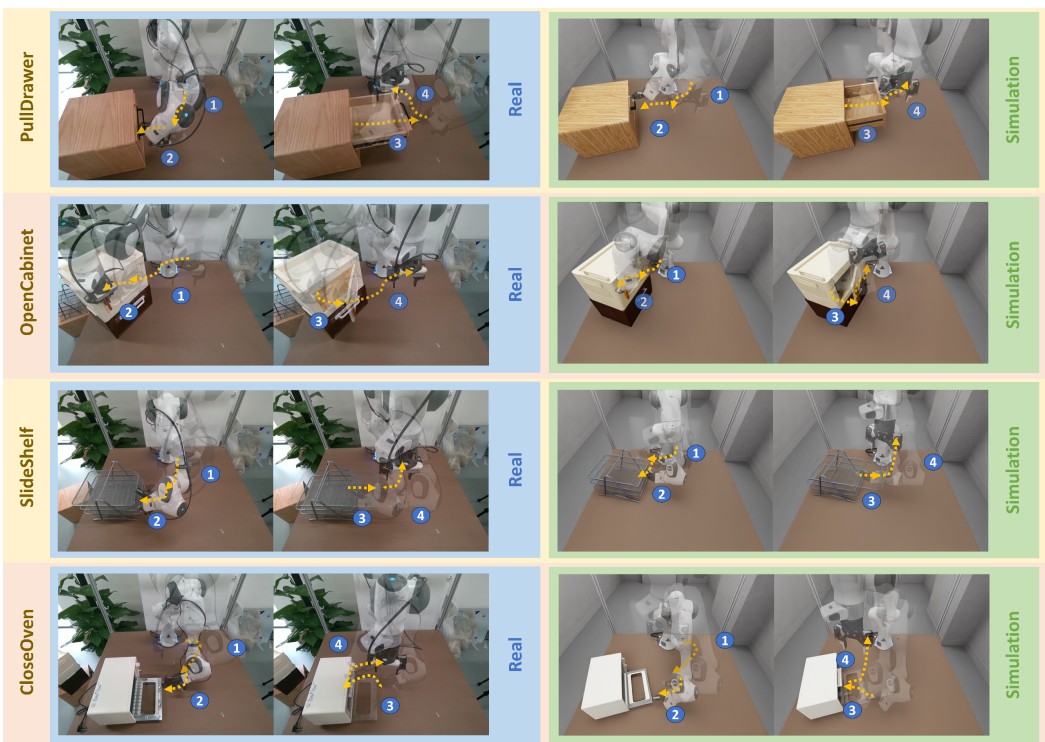

Figure 14: The four articulated-object manipulation tasks conducted for imitation learning.

| | Hyperparameter | Value | | Hyperparameter | Value |
|---|---|---|---|---|---|
| | Batch size | 48 | | Encoder layer | 4 |
| | Learning rate | 1e-4 | Network | Decoder layer | 7 |
| Training | Optimizer | AdamW | Architectures | Forward dim | 3200 |
| | KL weight | 10 | | Heads num | 8 |
| | Action sequence | 50 | | Transformer hidden dim | 512 |
| | Training step | 50k | | Backbone | ResNet50 |

Table 8: Implementation details of Action Chunking Transformer (ACT).

**Imitation Learning Algorithm.** The input to the imitation learning models consists of RGB image data from multiple camera views and the robot's proprioceptive states. The output is the robot control signals, such as joint positions, enabling end-to-end task execution. We used two state-of-the-art imitation learning methods, Action Chunking Transformer (ACT) (Zhao et al., 2023) and Diffusion Policy (DP) (Chi et al., 2023), to train the robotic policies for the articulated object manipulation task. Hyperparameters of both methods are demonstrated in Tab. 8 and Tab. 9.

- **Action Chunking Transformer (ACT)** (Zhao et al., 2023): ACT is built on the transformer network architecture and leverages temporal ensemble techniques to produce fluid and precise action sequences.
- **Diffusion Policy (DP)** (Chi et al., 2023): DP employs a diffusion-based generative model that captures multi-modal action distributions, offering robustness and high success rates for complex robotic tasks.

**Detailed Experiment Results.** The Full experiment results are presented in Tab. 10.

| | Hyperparameter | Value | | Hyperparameter | Value |
|---|---|---|---|---|---|
| | Batch size | 48 | | Diffuion Network | Unet1D |
| | Learning rate | 1e-4 | Network | Pooling | SpatialSoftmax |
| Training | Optimizer | AdamW | Architectures | Noise scheduler | DDIM |
| | EMA power | 0.75 | | EMA model | True |
| | Action sequence | 16 | | Noise schedule | SquaredcosCap |
| | Training step | 50k | | Backbone | ResNet50 |

Table 9: Implementation details of Diffusion Policy (DP).

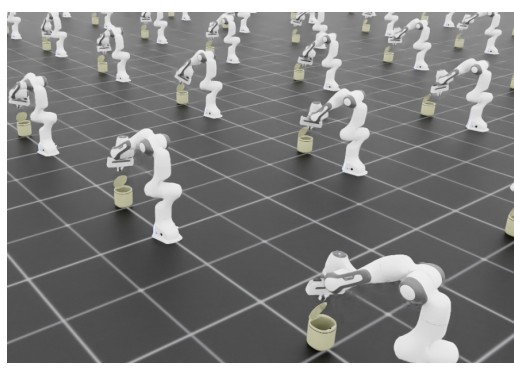

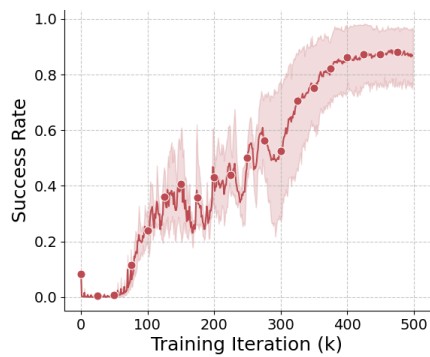

(a) Training task.          (b) Training curve over five random seeds.

Figure 15: RL-based training of visuomotor policy with ArtVIP.

## J    REINFORCEMENT LEARNING APPLICATION

**Training Details.** We extend the visual RL framework EAGLE (Zhao et al., 2025) to articulated-object tasks in ArtVIP. Fig. 15a shows the CloseTrashcan task, where the robot arm is required to close the trashcan within a given time limit. EAGLE is a two-stage visual RL framework designed for efficiency and generalization. In Stage 1, the teacher policy receives low-level states, including the robot arm's proprioceptive input, the lid's joint value, and the 3D relative position between the trashbin and the gripper. In Stage 2, the student policy is provided only with the wrist camera image and the robot's proprioceptive state—no object-related states are available. Fig. 15b presents the training curves in Stage 2.

For implementation details, in Stage 1, we replace EAGLE's original RL agent with PPO; In Stage 2, a privileged-state teacher is distilled into a visuomotor student while a self-supervised attention mask learned as follows:

$$\mathcal{L}_{att} = \mathcal{L}_{rec} + \mathcal{L}_{ae} + \beta\mathcal{L}_{ctl} + \lambda\mathcal{L}_{sps}, \tag{6}$$

where $\mathcal{L}_{rec}$ and $\mathcal{L}_{ae}$ are reconstruction losses, $\mathcal{L}_{ctl}$ predicts dynamics, and $\mathcal{L}_{sps}$ enforces mask sparsity. Hyper-parameters $\beta$ and $\lambda$ weight auxiliary losses.

The student policy is trained with the distillation loss:

$$\hat{\mathcal{L}}(\pi_\theta) = \mathbb{E}_{(\mathbf{o},\mathbf{s})\sim\mathcal{D}}\left[\|\pi_\theta(\mathbf{o}_{\text{aug}}) - \pi_e(\mathbf{s})\|_2^2\right], \tag{7}$$

where $\mathbf{s}$ contains privileged states and $\mathbf{o}_{\text{aug}}$ are images augmented by the learned mask with Eqn. equation 6. Hyper-parameters used in EAGLE are listed in Tab. 11.

**Reward Functions.** The **CloseTrashcan** task is a long-horizon challenge requiring the robot to first approach the trashcan lid and then close it smoothly. To facilitate efficient RL training, we design a multi-objective reward function as follows:

$$r_t(\boldsymbol{s}_t, \boldsymbol{a}_t) = \lambda_1 r_{dst}(\boldsymbol{s}_t) + \lambda_2 r_{dir}(\boldsymbol{s}_t) + \lambda_3 r_{cls}(\boldsymbol{s}_t) + \lambda_4 r_{smth}(\boldsymbol{a}_t), \tag{8}$$

where $r_{dst}$ rewards proximity between the gripper and the lid, $r_{dir}$ encourages alignment toward the lid, $r_{cls}$ measures lid closure progress, and $r_{smth}$ promotes smooth actions. The reward weights are set as: $\lambda_1 = 0.5$, $\lambda_2 = 0.125$, $\lambda_3 = 10$, $\lambda_4 = -0.01$.

Table 10: Performance results (scheme A): per-seed scores and mean $\pm$ 90% CI

| Task | Method | Strategy | Seed 1 | Seed 2 | Seed 3 | Mean $\pm$ CI$_{90}$ |
|---|---|---|---|---|---|---|
| PullDrawer | ACT | RO | 0.567 | 0.767 | 0.600 | 0.644 $\pm$ 0.059 |
| | | SO | 0.433 | 0.433 | 0.300 | 0.389 $\pm$ 0.060 |
| | | RSM100+10 | 0.500 | 0.667 | 0.767 | 0.640 $\pm$ 0.059 |
| | | RSM100+20 | 0.667 | 0.600 | 0.767 | 0.678 $\pm$ 0.057 |
| | | RSM100+50 | 0.833 | 0.767 | 0.733 | 0.778 $\pm$ 0.051 |
| | | RSM100+100 | 0.767 | 0.867 | 0.800 | 0.811 $\pm$ 0.048 |
| | DP | RO | 0.600 | 0.733 | 0.633 | 0.656 $\pm$ 0.058 |
| | | SO | 0.133 | 0.233 | 0.233 | 0.200 $\pm$ 0.049 |
| | | RSM100+10 | 0.600 | 0.650 | 0.700 | 0.650 $\pm$ 0.057 |
| | | RSM100+20 | 0.650 | 0.700 | 0.733 | 0.694 $\pm$ 0.056 |
| | | RSM100+50 | 0.700 | 0.733 | 0.750 | 0.728 $\pm$ 0.055 |
| | | RSM100+100 | 0.733 | 0.767 | 0.867 | 0.789 $\pm$ 0.050 |
| OpenCabinet | ACT | RO | 0.300 | 0.400 | 0.333 | 0.344 $\pm$ 0.058 |
| | | SO | 0.167 | 0.100 | 0.100 | 0.122 $\pm$ 0.040 |
| | | RSM100+10 | 0.333 | 0.367 | 0.367 | 0.356 $\pm$ 0.059 |
| | | RSM100+20 | 0.367 | 0.400 | 0.367 | 0.378 $\pm$ 0.059 |
| | | RSM100+50 | 0.433 | 0.500 | 0.400 | 0.444 $\pm$ 0.061 |
| | | RSM100+100 | 0.567 | 0.367 | 0.433 | 0.456 $\pm$ 0.061 |
| | DP | RO | 0.467 | 0.500 | 0.500 | 0.489 $\pm$ 0.061 |
| | | SO | 0.133 | 0.033 | 0.133 | 0.100 $\pm$ 0.037 |
| | | RSM100+10 | 0.500 | 0.533 | 0.567 | 0.533 $\pm$ 0.058 |
| | | RSM100+20 | 0.550 | 0.583 | 0.600 | 0.578 $\pm$ 0.057 |
| | | RSM100+50 | 0.600 | 0.617 | 0.633 | 0.617 $\pm$ 0.057 |
| | | RSM100+100 | 0.667 | 0.700 | 0.600 | 0.656 $\pm$ 0.058 |
| SlideShelf | ACT | RO | 0.233 | 0.233 | 0.333 | 0.267 $\pm$ 0.054 |
| | | SO | 0.100 | 0.167 | 0.133 | 0.133 $\pm$ 0.042 |
| | | RSM100+10 | 0.200 | 0.267 | 0.300 | 0.256 $\pm$ 0.053 |
| | | RSM100+20 | 0.233 | 0.300 | 0.267 | 0.267 $\pm$ 0.054 |
| | | RSM100+50 | 0.300 | 0.367 | 0.300 | 0.322 $\pm$ 0.057 |
| | | RSM100+100 | 0.333 | 0.333 | 0.400 | 0.356 $\pm$ 0.059 |
| | DP | RO | 0.467 | 0.433 | 0.433 | 0.444 $\pm$ 0.061 |
| | | SO | 0.167 | 0.167 | 0.200 | 0.178 $\pm$ 0.047 |
| | | RSM100+10 | 0.433 | 0.467 | 0.500 | 0.467 $\pm$ 0.058 |
| | | RSM100+20 | 0.500 | 0.533 | 0.550 | 0.528 $\pm$ 0.057 |
| | | RSM100+50 | 0.533 | 0.567 | 0.583 | 0.561 $\pm$ 0.056 |
| | | RSM100+100 | 0.567 | 0.600 | 0.600 | 0.589 $\pm$ 0.060 |
| CloseOven | ACT | RO | 0.500 | 0.633 | 0.600 | 0.578 $\pm$ 0.061 |
| | | SO | 0.267 | 0.267 | 0.167 | 0.233 $\pm$ 0.052 |
| | | RSM100+10 | 0.500 | 0.600 | 0.667 | 0.589 $\pm$ 0.060 |
| | | RSM100+20 | 0.533 | 0.633 | 0.633 | 0.600 $\pm$ 0.060 |
| | | RSM100+50 | 0.733 | 0.533 | 0.700 | 0.656 $\pm$ 0.058 |
| | | RSM100+100 | 0.667 | 0.800 | 0.567 | 0.678 $\pm$ 0.057 |
| | DP | RO | 0.600 | 0.700 | 0.667 | 0.656 $\pm$ 0.058 |
| | | SO | 0.267 | 0.233 | 0.333 | 0.278 $\pm$ 0.055 |
| | | RSM100+10 | 0.633 | 0.667 | 0.700 | 0.667 $\pm$ 0.057 |
| | | RSM100+20 | 0.667 | 0.700 | 0.733 | 0.700 $\pm$ 0.056 |
| | | RSM100+50 | 0.700 | 0.733 | 0.750 | 0.728 $\pm$ 0.055 |
| | | RSM100+100 | 0.767 | 0.733 | 0.833 | 0.778 $\pm$ 0.051 |

**Baseline Comparison.** To put EAGLE's performance in context, we compare it with a vision-based PPO method. As shown in Tab. 12, due to the high computation complexity and low data diversity, the baseline performs poorly on the CloseTrashcan task, while EAGLE achieves a 98% success rate after 500k training iterations.

|  | Hyperparameter | Value |
|---|---|---|
| Teacher (Stage 1) | Learning rate for all net | 5e-4 |
|  | Optimizer | Adam |
|  | Batch size | $12 \times 4096$ |
|  | Discount factor | 0.99 |
|  | Clip ratio | 0.2 |
|  | Rollout size | $96 \times 4096$ |
| Student (Stage 2) | Observation | $128 \times 128$ |
|  | Learning rate for all net | 1e-4 |
|  | Optimizer | Adam |
|  | Batch size | 256 |
|  | Frame stack | 1 |
|  | Replay buffer size | 100k |
|  | $\lambda$ | 0.01 |
|  | $\beta$ | 0.5 |
|  | $\alpha$ in *random overlay* | linear schedule from 0.4 to 0.9 |

Table 11: Hyperparamters for EAGLE.

Table 12: EAGLE vs. vision-based PPO: success rate across training checkpoints (k).

| Method | Training Iterations (k) | | | | |
|---|---|---|---|---|---|
|  | 100 | 200 | 300 | 400 | 500 |
| EAGLE | 0.23 | 0.28 | 0.73 | 0.85 | 0.98 |
| Vision-based PPO | 0.16 | 0.19 | 0.21 | 0.22 | 0.24 |

## K PEARSON CORRELATION COEFFICIENT DETAILS

Following (Li et al., 2024b), we compute the Pearson correlation coefficient from the success rates in Tab. 3 as

$$r = \frac{\sum_{i=1}^{n}(x_i - \bar{x})(y_i - \bar{y})}{\sqrt{\sum_{i=1}^{n}(x_i - \bar{x})^2}\sqrt{\sum_{i=1}^{n}(y_i - \bar{y})^2}},$$

where $x_i$ and $y_i$ denote the corresponding success rates in simulation and the real world at the $i$-th checkpoint. A high Pearson correlation indicates a strong linear relationship between simulated and real-world performance. The value $r = 0.9886$ using data from Tab. 3 shows that ArtVIP provides a reliable simulated training and evaluation pipeline for RL.

## L COMPARISON TO GENERATIVE PIPELINES

To evaluate the quality of generated assets, we reproduced SplArt (Lin et al., 2025) and generated a two–drawer cabinet and a side–by–side fridge. As shown in Fig. 16, the outputs are of lower quality than our digital–twin assets. Representative failures include 1) self-collisions, 2) severe mesh distortions and breakage, 3) incorrect joint limits, positions, and axes, 4) materials and colors that deviate markedly from reality, and 5) severe lack of interior details. Consequently, current generative baselines fail to produce simulation–ready articulated assets.

To quantify the domain gap, we evaluate reconstruction metrics on two real–world objects (cabinet and fridge) and compare them with the metrics reported by SplArt (Lin et al., 2025) for the same categories on synthetic data (Tab. 13). While SplArt achieves low errors on synthetic inputs, performance degrades markedly on real–world data, highlighting a substantial sim–to–real gap in articulated reconstruction. Additionally, the generated cabinet and fridge contain $\sim 100\times$ more triangles than their ArtVIP counterparts, reducing the simulation frame rate from $\sim 90$ fps to $\sim 70$ fps.

Limitations of generative pipelines: Many baselines (Lin et al., 2025; Mandi et al., 2024; Liu et al., 2025; Guo et al., 2025; Liu et al., 2024a; Su et al., 2024) are primarily trained and evaluated on synthetic data, where calibrated camera poses and controllable materials reduce domain gap. When performing inference on real images, they face:

1. Reconstruction accuracy degrades when camera extrinsics are estimated rather than known.

2. Since these methods are trained on PartNet–Mobility, generalization is constrained by that dataset's category coverage and articulation priors.

3. Flat surfaces and right–angle structures often appear wavy or warped under monocular/low–texture settings (e.g., Real2Code (Mandi et al., 2024), Fig. 5).

4. Reconstruction outputs frequently lack physically consistent materials and textures. 3DGS methods encode color via spherical harmonics (f_dc/f_rest) requiring custom shaders, so rendering–pipeline simulators (Isaac Sim, Sapien, PyBullet, MuJoCo, etc.) cannot render colors correctly without conversion/baking.

5. Current metrics do not capture mesh/triangle efficiency or collision performance relevant to simulation FPS.

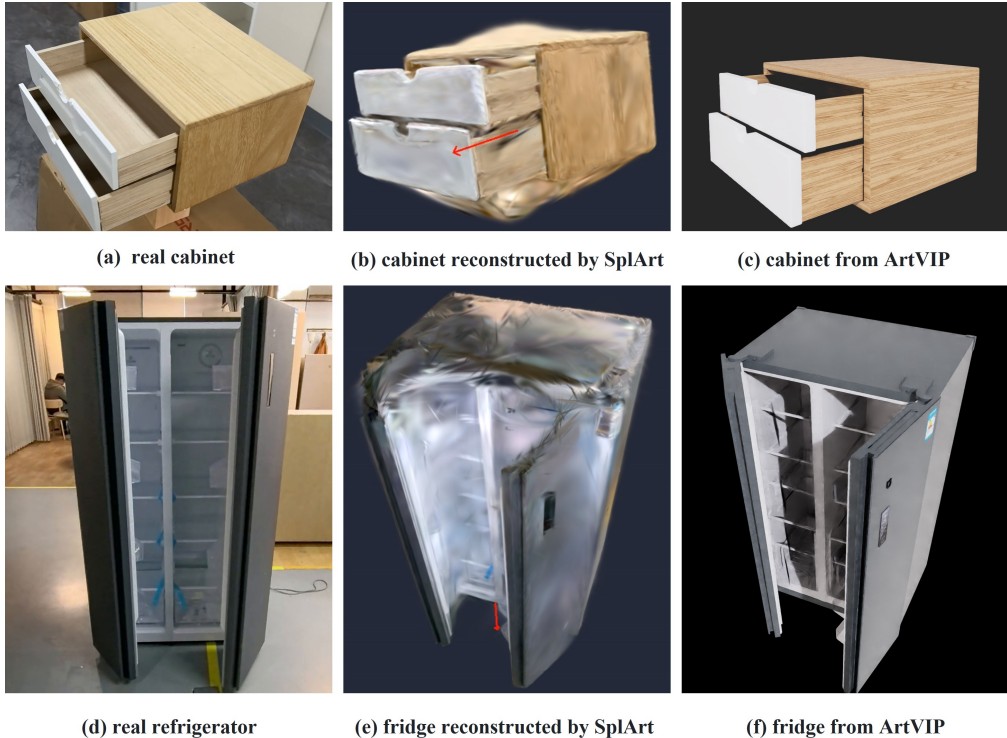

(a) real cabinet     (b) cabinet reconstructed by SplArt     (c) cabinet from ArtVIP

(d) real refrigerator     (e) fridge reconstructed by SplArt     (f) fridge from ArtVIP

Figure 16: Comparison of real-world objects, generated outputs, and digital-twin assets.

Table 13: Real-world vs. SplArt (synthetic) reconstruction metrics for cabinet and fridge.

| Metric | Real-World | | SplArt (Synthetic) | |
|---|---|---|---|---|
| | Cabinet | Fridge | Cabinet | Fridge |
| CD-Static | 13.71 | 11.04 | 7.31 | 0.52 |
| CD-Mobile | 11.50 | 13.45 | 1.02 | 0.27 |
| CD-Whole | 12.35 | 12.78 | 5.21 | 0.70 |
| Axis Ang. | 20.90 | N/A | 0.01 | 0.03 |
| Axis Pos. | 0.094 | N/A | – | 0.00 |

Notes: CD denotes Chamfer Distance (↓ better). Axis Angular Error and Axis Positional Error are of the joint axis (↓ better). "N/A" indicates the axis estimate cannot be produced (e.g., joint not detected).

## M  THE USE OF LARGE LANGUAGE MODELS.

A large language model (LLM) was used strictly as a writing aid for language polishing (grammar, clarity, and style). All ideas, methodological designs, datasets, code, analyses, and results are original and solely produced by the authors.

