# OpenReview forum: "ArtVIP: Articulated Digital Assets of Visual Realism, Modular Interaction, and Physical Fidelity for Robot Learning"
_ICLR.cc/2026/Conference — ICLR 2026 Poster_

### Official Review · Reviewer_CLY8 · 2025-10-29

**Soundness:** 3
**Presentation:** 4
**Contribution:** 3
**Rating:** 6
**Confidence:** 4

**Summary:**

This paper introduces ArtVIP, a novel open-source dataset of articulated digital assets designed for robot learning. It tackles the shortcomings of existing datasets by providing high visual realism, modular interaction capabilities, and physical fidelity, all within the USD format. The dataset includes 206 articulated objects across 26 categories, along with complementary indoor scene assets. The authors demonstrate its effectiveness through evaluations of visual and physical properties and validate its utility in imitation learning and reinforcement learning tasks, showing improved sim-to-real transfer and robust policy training.

**Strengths:**

1. This work offers a substantial collection of 206 articulated objects and scene assets, meticulously crafted by professional 3D modelers. The emphasis on visual realism (meshes, textures, PBR materials) and physical fidelity (collision, optimized joints) is a significant improvement over existing datasets.
2. The embedding of customizable, reusable behaviors directly within assets is a novel and highly beneficial feature. This greatly simplifies the development of interactive functionalities in simulations, reducing overhead and accelerating research.
3. The paper provides objective evaluation methods for both visual realism (triangular faces, reconstruction performance, feature distribution visualization with t-SNE and CLIP) and physical fidelity (optical motion capture comparing real-world vs. simulated joint movements under force). This rigorous evaluation adds significant credibility to the claims of high fidelity.
4. The extensive experiments in imitation learning (ACT, DP) and reinforcement learning (EAGLE framework) clearly demonstrate the practical utility of ArtVIP. The results showing zero-shot deployment capability and improved performance with mixed real-sim data are particularly compelling.

**Weaknesses:**

1.  While the paper mentions streamlined processes and scripting tools, the asset creation remains a manual process by professional 3D modelers. The paper acknowledges this as a limitation, stating, "scaling to even larger datasets remains a non-trivial challenge." This could be a significant barrier to expanding the dataset's diversity and size in the future without further automation.
2.  The enhancement of the joint drive equation, especially the functions of $q$ and $\dot{q}$, is crucial for physical fidelity. While the appendix provides the equations, a more detailed explanation of the derivation or the empirical process for determining parameters like $\mu_s$, $k_{high}$, $k_{low}$, $\alpha$, and $\lambda$ would strengthen this section.
3.  While the RL experiments validate the dataset, they focus on a single CloseTrashcan task. Expanding the range of RL tasks to demonstrate the utility of ArtVIP across more diverse manipulation challenges would further strengthen the claims.

Typo: Line 431: ACT 39% on PullDrawer

**Questions:**

1. Regarding the Modular Interaction feature: Can the authors provide more technical details or an example of how a behavior like $\textit{toggle door}$ is embedded directly into the assets in USD format without writing additional Python scripts? What is the underlying mechanism for this modularity and reusability?
2. The paper mentions the trade-off between the number of triangular faces and simulation frame rate (Appendix G). How did the authors balance these aspects, and what specific optimization techniques were employed beyond merging redundant vertices to ensure a consistent frame rate above 60 Hz across diverse scenes and objects?
3. In the imitation learning experiments (Tab. 1), the "Sim-Only" models consistently performed worse than "Real-Only" models, even with equal data quantity. While the paper attributes this to persistent challenges in bridging the sim-to-real gap, could the authors elaborate on specific aspects of the remaining gap (e.g., visual discrepancies, unmodeled physical phenomena, or limitations in the teleoperation data collection) that still contribute to this performance difference?
4. Given the acknowledged challenge of scaling asset creation, what concrete steps are planned for future work on generative approaches (e.g., specific generative models, data requirements, or methodologies) to automate asset synthesis and broaden diversity?

---

> ### Author Response · Authors · 2025-11-27
>
> We thank the reviewer for the valuable feedback. Below, we provide detailed responses to the reviewer's primary comments and concerns.
> ### Q1: Regarding the Modular Interaction feature: Can the authors provide more technical details or an example of how a behavior like  is embedded directly into the assets in USD format without writing additional Python scripts? What is the underlying mechanism for this modularity and reusability?
>
> The following pseudocode reproduces all joint behaviors of the Microwave.
> ```python
> class MicrowaveOverControl:
>   on_init():
>     set physics_frequency = 120
>     stiffness_max = 0.8
>     stiffness_min = 0.08
>     door_locked = true
>     cache local pose of button
>     get articulation by root path
>     get angular drive for revolute joint by path
>
>   on_play():
>     create SimulationContext
>
>   on_stop():
>     init_start = true
>     door_locked = true
>
>   on_update(current_time, delta_time):
>     if delta_time <= 0: return
>     acquire dynamic control
>     activate_articulation()
>     apply_behavior()
>
>   activate_articulation():
>     get articulation by root path
>     wake up articulation
>     joint_handles = [find_dof_handle(name) for name in joint_names]
>
>   apply_behavior():
>     oven_control()
>
> ```
> - modularity
>  We reference this behavior script inside the Microwave USD asset. When users load this USD directly in any Isaac Sim script (e.g., a benchmark using ArtVIP), the behavior logic executes automatically and correctly, fully decoupling benchmark code from asset joint-control logic.
> - reusability
> The same behavior script can be reused across microwave assets by repointing them to this script.

---

> > ### Author Response · Authors · 2025-11-27
> >
> > ### Q2: The paper mentions the trade-off between the number of triangular faces and simulation frame rate (Appendix G). How did the authors balance these aspects, and what specific optimization techniques were employed beyond merging redundant vertices to ensure a consistent frame rate above 60 Hz across diverse scenes and objects?
> > We analyze the relationship between triangle count and FPS and provide optimization guidelines.
> >
> > To study the effect of triangle count, we report comprehensive statistics in Tab.7 (Appendix) for ArtVIP, PartNet-Mobility, and BEHAVIOR-1K: the average triangle count, the average number of active joints, the average FPS with a single asset, and the average FPS in the kitchen scene. The kitchen scene is the most complex environment, containing 65 actuated joints. The evaluation results are as follows.
> >
> > | Item Category   | Single item (ArtVIP) | Single item (PartNet) | Single item (B1K) | Item in Kitchen (ArtVIP) | Item in Kitchen (PartNet) | Item in Kitchen (B1K) |
> > |-----------------|----------------:|---------------------------:|---------------------:|-----------------:|----------------------------:|----------------------:|
> > | Coffee Machine  | 91.97           | 91.95                      | 114.77               | 72.01            | 73.83                       | 48.33                 |
> > | Microwave       | 91.93           | 91.88                      | 87.51                | 69.38            | 72.95                       | 50.04                 |
> > | Oven            | 91.98           | 91.95                      | 109.97               | 72.66            | 74.64                       | 49.40                 |
> > | Dishwasher      | 91.94           | 91.95                      | 115.20               | 65.75            | 69.95                       | 49.73                 |
> > | Rice Cooker     | 91.96           | 91.96                      | 115.58               | 70.97            | 74.91                       | 48.29                 |
> > | Laptop          | 91.97           | 91.93                      | 112.00               | 74.59            | 74.89                       | 46.99                 |
> > | Washing Machine | 91.95           | 91.94                      | 107.17               | 70.97            | 70.74                       | 48.27                 |
> > | Toilet          | 91.95           | 91.95                      | 120.58               | 74.94            | 74.93                       | 47.18                 |
> > | Refrigerator    | 91.94           | 91.96                      | 99.55                | 60.78            | 73.89                       | 49.82                 |
> > | Table           | 91.96           | 91.96                      | 116.37               | 68.13            | 71.68                       | 47.92                 |
> > | Folding Chair   | 91.95           | 91.97                      | 125.93               | 74.50            | 74.92                       | 51.41                 |
> > | Scissors        | 91.96           | 91.96                      | 129.71               | 72.50            | 74.92                       | 52.45                 |
> > | Trash Can       | 91.94           | 91.93                      | 121.28               | 71.66            | 74.93                       | 52.24                 |
> >
> > Based on the FPS results for ArtVIP and PartNet-Mobility, we conclude:
> > - For IL/RL, an FPS ideally above 60 is needed to match camera sampling frequency, but BEHAVIOR‑1K via OmniGibson does not meet this requirement.
> > - For a single object, under Isaac Sim's iterative optimizations, triangle counts up to approximately 100k and up to 20 active joints have negligible impact on FPS.
> > - In complex scenes, both triangle count and the number of active joints reduce FPS.
> >
> > Many factors affect simulation FPS, including polygon count, collision‑shape optimization, counts of static meshes and active rigid bodies, textures and materials, etc. Their impact is complicated; maintaining stable FPS therefore depends heavily on simulation engineering practices. Below are several practical optimizations that are not officially documented in Isaac Sim but have a large impact and implemented in ArtVIP:
> > 1. In articulated objects, minimize the number of active rigid‑body parts and use static meshes wherever parts never move. For example, cabinet tops and side panels are static, exclude them from kinematics calculation to improve simulation efficiency.
> > 2. Optimize collision geometry: use primitive shapes (e.g., cubes, cylinders) to replace complex collision meshes. This significantly improves collision‑detection efficiency and reduces computation.

---

> > > ### Author Response · Authors · 2025-11-27
> > >
> > > ### Q3: elaborate on specific aspects of the remaining gap (e.g., visual discrepancies, unmodeled physical phenomena, or limitations in the teleoperation data collection) that still contribute to this performance difference?
> > >
> > > Visual discrepancies contribute to this performance difference.
> > > As shown in Fig. 5, we visualize this gap and demonstrate that professional 3d modeling reduces the sim-to-real gap. In real-world experiments, imitation-learning methods such as ACT and DP are highly vision-dependent; zero-shot Sim-Only results are poor, which further confirms that the remaining gap currently lies in vision.
> > > We also added an experiment in Sec. 5.2: a cross‑dataset comparison against PartNet‑Mobility, which is the source of articulated objects used in RoboTwin. We kept the original background unchanged. We selected five microwave ovens from PartNet‑Mobility and five from ArtVIP, and for the real‑world task we purchased an unseen microwave oven. The results are summarized in the table below. We found that higher‑fidelity ArtVIP assets yield stronger zero‑shot sim‑to‑real transfer and higher success under Real–Sim–Mixed settings, supporting the conclusion that higher‑quality assets reduce the sim‑to‑real gap and lead to higher success rates.
> > >
> > > | Method | Dataset    | Ours | PartNet-Mobility |
> > > |--------|------------|-----:|-----------------:|
> > > | ACT    | RO         | 56%   | 56%              |
> > > | ACT    | SO         | 41% (+9 pp)  | 32%              |
> > > | ACT    | RSM100+500 | 79% (+11 pp) | 68%              |
> > > | DP     | RO         | 62%   | 62%              |
> > > | DP     | SO         | 45% (+10 pp) | 35%              |
> > > | DP     | RSM100+500 | 83% (+13 pp) | 70%              |

---

> > > > ### Author Response · Authors · 2025-11-27
> > > >
> > > > ### Q4: Given the acknowledged challenge of scaling asset creation, what concrete steps are planned for future work on generative approaches (e.g., specific generative models, data requirements, or methodologies) to automate asset synthesis and broaden diversity?
> > > > During the development of ArtVIP, we summarized methods that streamline workflow and reduce costs, and applied them in recent production runs, yielding promising results.
> > > > We have not integrated generative pipelines because the generated assets are low‑quality and often cause simulation errors at the current stage.
> > > >
> > > > #### 1.Scaling via principled optimizations:
> > > > During the development of ArtVIP, after producing the first ~500 assets, the average per‑asset time falls by ~20–30%, and after ~1,000 assets it declines by ~50–60%, showing efficient dataset scaling. These gains are driven by:
> > > > - Built a reusable library of 3D parts and textures. As the library expands, modeling time continuously decreases.
> > > > - Modularized interaction components and auto‑generated scripts for physical parameter tuning, reducing tuning time by ~90%.
> > > >
> > > > #### 2.Generative pipelines can't be integrated into simulation workflows:
> > > > We reproduced SplArt [1] and generated assets for a two‑drawer cabinet and a side‑by‑side refrigerator. The resulting outputs are low‑quality and trigger runtime errors in Isaac Sim. Consequently, current generative baselines fail to produce simulation‑ready articulated assets for robotics. The process and output assets are available here. Image comparison is provided in Appendix Fig. 16.
> > > > Process: [SplArt Experiment Records](https://github.com/UejnYaM/generative_pipelines_reproduce/blob/main/sqlgs/SplArt%20Experiment%20Records.md)
> > > > Generated assets: [Cabinet (mp4)](https://github.com/UejnYaM/generative_pipelines_reproduce/blob/main/sqlgs/output/cabinet.mp4), [Fridge (mp4)](https://github.com/UejnYaM/generative_pipelines_reproduce/blob/main/sqlgs/output/fridge.mp4)
> > > >
> > > > From the above videos, the generated assets exhibit the issues of 1) self‑collisions, 2) severe mesh distortions and breakage, 3) incorrect joint limits, positions, and axes, 4) materials and colors that deviate markedly from reality, and 5) severe lack of interior details.
> > > >
> > > > We compare commonly used metrics for two real‑world objects (cabinet, fridge) against synthetic results reported in Tab.1 of ArtGS [3] to illustrate domain gap and failure cases in articulated reconstruction. Additionally, the generated cabinet and fridge contain ~100× more triangles than their ArtVIP counterparts, reducing the simulation frame rate from ~90 fps to ~70 fps.
> > > >
> > > > | Metric    | Real Cabinet | Real Fridge | Synthetic (ArtGS) Cabinet | Synthetic (ArtGS) Fridge |
> > > > |-----------|--------:|-------:|:-----------|:----------|
> > > > | CD-static |  13.71  |  11.04 | 7.31 | 0.52 |
> > > > | CD-mobile |  11.50  |  13.45 | 1.02 | 0.27 |
> > > > | CD-whole  |  12.35  |  12.78 | 5.21 | 0.7 |
> > > > | Axis Ang  |  20.90  |  N/A | 0.01 |0.03 |
> > > > | Axis Pos  |  0.094  |  N/A | - | 0.00 |
> > > >
> > > > Notes: CD denotes Chamfer Distance (↓ better). Axis Angular Error and Axis Positional Error are of the joint axis (↓ better). "N/A" indicates the axis estimate can not be produced (e.g., joint not detected).
> > > >
> > > > #### 3.Limitations of generative pipelines:
> > > > Many baselines [1~6] are primarily trained and evaluated on synthetic data, where calibrated camera poses and controllable materials reduce domain gap. When performing inference on real images, they face:
> > > > 1. Reconstruction accuracy degrades when camera extrinsics are estimated rather than known.
> > > > 2. Since [1~6] are trained on PartNet‑Mobility, generalization is constrained by that dataset’s category coverage and articulation priors.
> > > > 3. Flat surfaces and right‑angle structures often appear wavy or warped under monocular/low‑texture settings. (e.g., Figure 5 in Real2Code[2]).
> > > > 4. Reconstruction outputs frequently lack physically consistent materials and textures. 3DGS methods encodes color via spherical harmonics (f_dc/f_rest) requiring custom shaders, so rendering pipeline simulators (Isaac Sim, Sapien, PyBullet, Mujoco, etc.) cannot render colors correctly without conversion/baking.
> > > > 5. Current metrics do not capture mesh/triangle efficiency or collision performance relevant to simulation FPS.

---

> > > > > ### Author Response · Authors · 2025-11-27
> > > > >
> > > > > ### W1: Scaling to even larger datasets remains a non-trivial challenge
> > > > > We could not locate the exact phrasing "scaling to even larger datasets remains a non‑trivial challenge" in the original paper, but the concern is valid. We provide a detailed response in Q1.
> > > > >
> > > > > ### W2: The enhancement of the joint drive equation
> > > > > PhysX in Isaac Sim does not support differentiable solving, so we cannot customize the solver as in MuJoCo. Therefore, we enumerate common situations that frequently occur in articulated objects and provide general solutions.
> > > > >
> > > > > ### W3: Expanding the range of RL tasks
> > > > > We added an experiment in Sec. 5.2: a cross‑dataset comparison against PartNet‑Mobility, which is the source of articulated objects used in RoboTwin. We kept the original background unchanged. We selected five microwave ovens from PartNet‑Mobility and five from ArtVIP, and for the real‑world task we purchased an unseen microwave oven. The results are summarized in the table below. We found that higher‑fidelity ArtVIP assets yield stronger zero‑shot sim‑to‑real transfer and higher success under Real–Sim–Mixed settings, supporting the conclusion that higher‑quality assets reduce the sim‑to‑real gap and lead to higher success rates.
> > > > >
> > > > > | Method | Dataset    | Ours | PartNet-Mobility |
> > > > > |--------|------------|-----:|-----------------:|
> > > > > | ACT    | RO         | 56%   | 56%              |
> > > > > | ACT    | SO         | 41% (+9 pp)  | 32%              |
> > > > > | ACT    | RSM100+500 | 79% (+11 pp) | 68%              |
> > > > > | DP     | RO         | 62%   | 62%              |
> > > > > | DP     | SO         | 45% (+10 pp) | 35%              |
> > > > > | DP     | RSM100+500 | 83% (+13 pp) | 70%              |
> > > > >
> > > > >
> > > > >
> > > > > Reference:
> > > > > [1] Lin et al. SplArt: Articulation Estimation and Part-Level Reconstruction with 3D Gaussian Splatting, ICCV 2025
> > > > > [2] Mandi et al. Real2Code: Reconstruct Articulated Objects via Code Generation, ICLR 2025
> > > > > [3] Liu, Yu, et al. "Artgs: Building interactable replicas of complex articulated objects via gaussian splatting."
> > > > > [4] Guo, Junfu, et al. "Articulatedgs: Self-supervised digital twin modeling of articulated objects using 3d gaussian splatting."
> > > > > [5] Liu, Jiayi, et al. "Singapo: Single image controlled generation of articulated parts in objects."
> > > > > [6] Su, Jiayi, et al. "Artformer: Controllable generation of diverse 3d articulated objects."

---

### Official Review · Reviewer_jg1J · 2025-10-31

**Soundness:** 3
**Presentation:** 3
**Contribution:** 2
**Rating:** 2
**Confidence:** 4

**Summary:**

This paper presents ArtVIP, an open-source dataset of 992 articulated digital-twin objects with a focus on visual realism, modular interaction, and physical fidelity. All assets are professionally modeled under unified geometric and material standards and provided in USD format for compatibility with Isaac Sim and other simulators. The dataset includes detailed annotations, embedded interaction primitives, and scene assets for direct use in robot learning. The authors evaluate ArtVIP through reconstruction and feature-distribution analyses, optical motion capture of joint dynamics, and downstream imitation and reinforcement learning tasks. The work aims to bridge the sim-to-real gap by offering high-quality, physically consistent assets and comprehensive guidelines for digital-twin creation.

**Strengths:**

The paper presents ArtVIP, a high-quality open-source dataset of articulated digital objects designed to enhance simulation fidelity for robot learning. Its key strength lies in the professional-level asset quality, with rigorous modeling standards ensuring both visual realism and physical fidelity. The dataset includes modular, interactive behaviors and pixel-level affordance annotations, enabling more complex manipulation tasks in simulation.

Comprehensive evaluations including feature-level realism, physical motion consistency, and sim-to-real experiments in both imitation and reinforcement learning, demonstrate its practical utility. ArtVIP significantly advances the state of datasets for embodied AI by addressing limitations in existing resources and supporting immediate deployment in high-fidelity simulators.

**Weaknesses:**

While ArtVIP presents a high-quality dataset with clear efforts toward visual and physical fidelity, the evaluation methodology lacks quantitative rigor and convincing evidence. For the Visual Realism evaluation, the qualitative comparison in Figure 5 does not convincingly demonstrate that OmniGibson is inferior in geometry or texture realism. The statement that “reconstructions from ArtVIP assets exhibit higher structural fidelity and finer detail preservation” is insufficiently supported: no quantitative reconstruction metrics (e.g., PSNR, SSIM, Chamfer distance) are reported, and the visual difference is subtle and subjective. Likewise, the CLIP-based t-SNE visualization provides only superficial evidence: the feature clusters for Sim-ArtVIP and real data in Figure 5 (right) still show limited overlap, making the claim of stronger alignment somewhat speculative. Moreover, the paper does not provide dataset statistics (e.g., per-category object count, distribution of joint types, or texture resolutions) in the main text, which makes it difficult to assess the dataset’s scale and diversity without consulting the appendix.

**Questions:**

For the Reinforcement Learning part, the interpretation of the Pearson correlation between simulation and real trials as evidence of “high physical fidelity and visual realism” is unconvincing. A high correlation merely indicates consistent performance ranking across environments, not necessarily that the simulated physics or visuals faithfully reproduce real-world behavior. The authors should clarify (1) how the correlation is computed: over what quantities and across how many runs; and (2) why training exclusively in simulation would not trivially yield a similar correlation pattern. Without such clarifications, the claimed sim-to-real validity remains ambiguous.

---

> ### Author Response · Authors · 2025-11-27
>
> We thank the reviewer for the valuable feedback. Below, we provide detailed responses to the reviewer's primary comments and concerns.
> ### Q1&Q2: How the correlation is computed: over what quantities and across how many runs.Why training exclusively in simulation would not trivially yield a similar correlation pattern.
>
> To clarify the RL training and evaluation pipeline, we will revise Lines 473–476 as follows:
>
> "We train the RL policy in simulation and then deploy it in the real world on the same task, using VisualMatching (Li et al., 2024b) to ensure sim–real visual consistency. Tab. 3 reports success rates at five checkpoints between 300k and 500k training iterations, each evaluated with 100 trials in simulation and 30 trials in the real world under diverse initial object poses. The RL policy trained in ArtVIP exhibits an absolute sim-to-real success-rate gap of 0.05, indicating that ArtVIP provides high physical fidelity and visual realism. Following (Li et al., 2024b), we compute the Pearson correlation coefficient from the success rates in Tab. 3 as
>
> $$r = \frac{\sum_{i=1}^n (x_i - \bar{x})(y_i - \bar{y})}{\sqrt{\sum_{i=1}^n (x_i - \bar{x})^2}\sqrt{\sum_{i=1}^n (y_i - \bar{y})^2}},$$
>
> where $x_i$ and $y_i$ denote the corresponding success rates in simulation and the real world at the $i$-th checkpoint. A high Pearson correlation indicates a strong linear relationship between simulated and real-world performance. The value $r = 0.9886$ using data from Tab. 3 shows that ArtVIP provides a reliable simulated training and evaluation pipeline for RL."
>
>
> Pearson correlation assesses the linear fit between real and simulated performance[1]. In [1], the average Pearson correlation across all tasks is 0.929. In [2], the average Pearson correlation across all tasks is 0.920. In [4], RL navigation reports Sim2Real correlation for success at 0.18 under default settings and 0.844 after simulator tuning, both below 0.98. These results show strong but non‑perfect alignment. Our r = 0.9886 is higher because the RL policy in ArtVIP is trained on diverse large‑scale data and leverages EAGLE [3] control‑aware augmentation that strengthens visual generalization, which reduces the sim‑to‑real gap.
>
> ### W1: Lack quantitative rigor and convincing evidence in Fig.5
> We have supplemented quantitative high-dimensional metrics in the same CLIP embedding space:
>
> | Metric | Real–OmniGibson | Real–ArtVIP |
> | --- | --- | --- |
> | Mean Cosine Distance (full pairwise) | 0.3895 | 0.3067 |
> | Mean Cosine Distance (nearest-neighbor matching) | 0.1742 | 0.1253 |
> | FID (unnormalized features) | 122.135 | 106.307 |
> | NNO (K=1, sample number=500) | 0.0 | 0.1 |
>
> ### W2：Reconstruction metrics in Fig.5
> Because BEHAVIOR‑1K (B1K) assets are encrypted and accessible only via OmniGibson, we cannot access ground‑truth PLY files and therefore cannot compute Chamfer Distance. PSNR and SSIM reported here indicate that "reconstructions from ArtVIP assets exhibit higher structural fidelity and finer detail preservation".
>
> |   | Reconstruction from B1K frame | Reconstruction from ArtVIP frame |
> | --- | --- | --- |
> | PSNR (↑) | 19.3 | 23.6 |
> | SSIM (↑) | 0.82 | 0.84 |
>
> ### W3: The paper does not provide dataset statistics in the main text.
> In the initial version, we provided references in lines L193–L195 to the statistics in the Appendix, because **the detailed statistics exceeded the space constraints of the main text**. Tab. 5 reports the per‑category object counts, and Tab. 6 summarizes the distribution of joint types.
>
> Reference:
> [1] Li, Xuanlin, et al. "Evaluating real-world robot manipulation policies in simulation."
> [2] Zhang, Kaifeng, et al. "Real-to-Sim Robot Policy Evaluation with Gaussian Splatting Simulation of Soft-Body Interactions."
> [3] Zhao, Yinuo, et al. "Efficient Training of Generalizable Visuomotor Policies via Control-Aware Augmentation."
> [4] Kadian, Abhishek, et al. "Sim2Real Predictivity: Does Evaluation in Simulation Predict Real-World Performance?"

---

### Official Review · Reviewer_YbiS · 2025-11-01

**Soundness:** 3
**Presentation:** 3
**Contribution:** 3
**Rating:** 8
**Confidence:** 3

**Summary:**

The paper introduces ArtVIP, a high-quality open-source dataset of articulated digital-twin assets designed for robotic learning.
It highlights the visual realism, physical fidelity and the modular interaction. Quantitative experiments validate the assets through reconstruction fidelity, optical motion-capture comparisons, imitation-learning and reinforcement-learning benchmarks.

**Strengths:**

1. Encoding interactive semantics (like damping or cross-asset effects) directly in the USD assets is novel and practically useful.
2. Zero-shot sim-to-real transfer in imitation learning demonstrate the benefits of the proposed dataset for robotics learning.
3. The paper is very well written and the analysis is comprehensive.

**Weaknesses:**

1. While fidelity is high, 992 objects are modest. Integration with generative pipelines would strengthen long-term impact.
2. The dataset emphasizes three key aspects: visual realism, physical fidelity, and modular interaction, but the paper lacks quantitative analysis demonstrating the individual contribution of each component. Conducting ablation studies that selectively degrade or remove each aspect and measuring the resulting impact on imitation-learning and reinforcement-learning policy performance would help substantiate the significance of these design dimensions and strengthen the paper’s empirical validation.
4. The results can be enhanced by comparing IL and RL policies learned on baseline datasets (BEHAVIOR).

**Questions:**

1. The modeling and tuning time is very long. I wondering for the modeling time, how many time is spent on geometry, visual texture and collision modeling respectively.
2. For RL, is the RL trained purely in sim and zero-shot generalize to real?

---

> ### Author Response · Authors · 2025-11-27
>
> We thank the reviewer for the valuable feedback. Below, we provide detailed responses to the reviewer's primary comments and concerns.
> ### Q1: The modeling and tuning time is very long. I wondering for the modeling time, how many time is spent on geometry, visual texture and collision modeling respectively.
> Modeling time comprises geometry, visual texturing, and collision modeling. Typically, about 20% of a model’s modeling time is allocated to collision modeling; the modeler selects appropriate collision settings (e.g., a convex hull) and, if necessary (see lines 234–241), splits complex collision volumes into multiple primitive meshes (e.g., cubes, cylinders). Geometry accounts for approximately 40–70% of the modeling time. For categories with complex curved surfaces but simple textures (e.g., toilets), geometry modeling time can be about 70% and visual texturing about 10%. For doors, where curved‑surface modeling is simpler, geometry modeling time is around 40%.
> Across the dataset (modeling time only), the estimated median split is approximately: geometry 55%, visual texturing 25%, collision modeling 20%.
>
> ### Q2: For RL, is the RL trained purely in sim and zero-shot generalize to real?
> **Yes, it is zero-shot.** The RL policy in ArtVIP is trained on diverse large‑scale data and leverages EAGLE [3] control‑aware augmentation that strengthens visual generalization, which reduces the sim‑to‑real gap.

---

> > ### Author Response · Authors · 2025-11-27
> >
> > ### W1: Integration with generative pipelines would strengthen long-term impact
> > During the development of ArtVIP, we summarized methods that streamline workflow and reduce costs, and applied them in recent production runs, yielding promising results.
> > We have not integrated generative pipelines because the generated assets are low‑quality and often cause simulation errors at the current stage.
> >
> > #### 1.Scaling via principled optimizations:
> > During the development of ArtVIP, after producing the first ~500 assets, the average per‑asset time falls by ~20–30%, and after ~1,000 assets it declines by ~50–60%, showing efficient dataset scaling. These gains are driven by:
> > - Built a reusable library of 3D parts and textures. As the library expands, modeling time continuously decreases.
> > - Modularized interaction components and auto‑generated scripts for physical parameter tuning, reducing tuning time by ~90%.
> >
> > #### 2.Generative pipelines can't be integrated into simulation workflows:
> > We reproduced SplArt [4] and generated assets for a two‑drawer cabinet and a side‑by‑side refrigerator. The resulting outputs are low‑quality and trigger runtime errors in Isaac Sim. Consequently, current generative baselines fail to produce simulation‑ready articulated assets for robotics. The process and output assets are available here. Image comparison is provided in Appendix Fig. 16.
> > Process: [SplArt Experiment Records](https://github.com/UejnYaM/generative_pipelines_reproduce/blob/main/sqlgs/SplArt%20Experiment%20Records.md)
> > Generated assets: [Cabinet (mp4)](https://github.com/UejnYaM/generative_pipelines_reproduce/blob/main/sqlgs/output/cabinet.mp4), [Fridge (mp4)](https://github.com/UejnYaM/generative_pipelines_reproduce/blob/main/sqlgs/output/fridge.mp4)
> >
> > From the above videos, the generated assets exhibit the issues of 1) self‑collisions, 2) severe mesh distortions and breakage, 3) incorrect joint limits, positions, and axes, 4) materials and colors that deviate markedly from reality, and 5) severe lack of interior details.
> >
> > We compare commonly used metrics for two real‑world objects (cabinet, fridge) against synthetic results reported in Tab.1 of ArtGS [6] to illustrate domain gap and failure cases in articulated reconstruction. Additionally, the generated cabinet and fridge contain ~100× more triangles than their ArtVIP counterparts, reducing the simulation frame rate from ~90 fps to ~70 fps.
> >
> > | Metric    | Real Cabinet | Real Fridge | Synthetic (ArtGS) Cabinet | Synthetic (ArtGS) Fridge |
> > |-----------|--------:|-------:|:-----------|:----------|
> > | CD-static |  13.71  |  11.04 | 7.31 | 0.52 |
> > | CD-mobile |  11.50  |  13.45 | 1.02 | 0.27 |
> > | CD-whole  |  12.35  |  12.78 | 5.21 | 0.7 |
> > | Axis Ang  |  20.90  |  N/A | 0.01 |0.03 |
> > | Axis Pos  |  0.094  |  N/A | - | 0.00 |
> >
> > Notes: CD denotes Chamfer Distance (↓ better). Axis Angular Error and Axis Positional Error are of the joint axis (↓ better). "N/A" indicates the axis estimate can not be produced (e.g., joint not detected).
> >
> > #### 3.Limitations of generative pipelines:
> > Many baselines [3~9] are primarily trained and evaluated on synthetic data, where calibrated camera poses and controllable materials reduce domain gap. When performing inference on real images, they face:
> > 1. Reconstruction accuracy degrades when camera extrinsics are estimated rather than known.
> > 2. Since [3~9] are trained on PartNet‑Mobility, generalization is constrained by that dataset’s category coverage and articulation priors.
> > 3. Flat surfaces and right‑angle structures often appear wavy or warped under monocular/low‑texture settings. (e.g., Figure 5 in Real2Code[5]).
> > 4. Reconstruction outputs frequently lack physically consistent materials and textures. 3DGS methods encodes color via spherical harmonics (f_dc/f_rest) requiring custom shaders, so rendering pipeline simulators (Isaac Sim, Sapien, PyBullet, Mujoco, etc.) cannot render colors correctly without conversion/baking.
> > 5. Current metrics do not capture mesh/triangle efficiency or collision performance relevant to simulation FPS.

---

> > > ### Author Response · Authors · 2025-11-27
> > >
> > > ### W2: The paper lacks quantitative analysis demonstrating the individual contribution of each component(visual realism, physical fidelity, and modular interaction).
> > > We conduct ablation experiments and report quantitative analyses demonstrating the individual contribution of each component. Each component contributes to the sim‑to‑real success rate.
> > >
> > > To isolate the individual contribution of visual realism, we remove all textures from the cabinet in the OpenCabinet task and bake materials using only base colors (brown handle, light-yellow body). We also replace all surface decorations with flat surfaces. Under this ablation, the success rate under the Sim‑Only (SO) setting drops from 12% to 3%.
> > >
> > > To isolate the individual contribution of physical fidelity, in PullDrawer, we remove the tuned physical parameters of the drawer’s prismatic joint and set the handle to default friction. Under the Sim‑Only (SO) setting, the success rate decreases from 39% to 17%.
> > >
> > > As for modular interaction, an ablation that disables the interaction module between the switch and the lamp results in the lamp no longer turning on or off when the switch is pressed, yielding an interaction success rate of 0%.
> > >
> > > | Component | Task        | Setting         | Before ablation | After ablation | Δ (pp) |
> > > | ---       | ---         | ---             | ---:            | ---:           | ---:   |
> > > | Visual realism    | OpenCabinet | Sim‑Only (SO)   | 12%             | 3%            | -9    |
> > > | Physical fidelity | PullDrawer  | Sim‑Only (SO)   | 39%             | 17%           | -22   |
> > > | Modular interaction | Switch‑controlled lamp on/off | Default        | 100%             | 0%            | -100     |
> > >
> > > ### W3: Comparing IL and RL policies learned on baseline datasets (BEHAVIOR)
> > > We added an experiment in Sec. 5.2: a cross‑dataset comparison against PartNet‑Mobility, which is the source of articulated objects used in RoboTwin. We kept the original background unchanged. We selected five microwave ovens from PartNet‑Mobility and five from ArtVIP, and for the real‑world task we purchased an unseen microwave oven. The results are summarized in the table below. We found that higher‑fidelity ArtVIP assets yield stronger zero‑shot sim‑to‑real transfer and higher success under Real–Sim–Mixed settings, supporting the conclusion that higher‑quality assets reduce the sim‑to‑real gap and lead to higher success rates.
> > >
> > > | Method | Dataset    | Ours | PartNet-Mobility |
> > > |--------|------------|-----:|-----------------:|
> > > | ACT    | RO         | 56%   | 56%              |
> > > | ACT    | SO         | 41% (+9 pp)  | 32%              |
> > > | ACT    | RSM100+500 | 79% (+11 pp) | 68%              |
> > > | DP     | RO         | 62%   | 62%              |
> > > | DP     | SO         | 45% (+10 pp) | 35%              |
> > > | DP     | RSM100+500 | 83% (+13 pp) | 70%              |
> > >
> > >
> > > Reference:
> > > [1] Lin et al. SplArt: Articulation Estimation and Part-Level Reconstruction with 3D Gaussian Splatting, ICCV 2025
> > > [2] https://behavior.stanford.edu/behavior_components/asset_sources.html
> > > [3] Xiang, Fanbo, et al. "Sapien: A simulated part-based interactive environment."
> > > [4] Lin et al. SplArt: Articulation Estimation and Part-Level Reconstruction with 3D Gaussian Splatting, ICCV 2025
> > > [5] Mandi et al. Real2Code: Reconstruct Articulated Objects via Code Generation, ICLR 2025
> > > [6] Liu, Yu, et al. "Artgs: Building interactable replicas of complex articulated objects via gaussian splatting."
> > > [7] Guo, Junfu, et al. "Articulatedgs: Self-supervised digital twin modeling of articulated objects using 3d gaussian splatting."
> > > [8] Liu, Jiayi, et al. "Singapo: Single image controlled generation of articulated parts in objects."
> > > [9] Su, Jiayi, et al. "Artformer: Controllable generation of diverse 3d articulated objects."

---

### Official Review · Reviewer_1Zfo · 2025-11-06

**Soundness:** 3
**Presentation:** 3
**Contribution:** 2
**Rating:** 6
**Confidence:** 4

**Summary:**

ArtVIP presents a dataset of almost 1000 digital-twin articulated objects in USD format, where all assets are visually realistic and exhibit physical fidelity. The dataset is manually crafted from 3D modeling experts following a given assembly guidelines. One key aspect of the paper is the interactive functionality of the different assets, where 5 key behavior primitives are integrated. This makes it possible to customize objects and still having those interaction functionalities, without the need of rewriting code. The dataset is evaluated on triangle count, reconstruction performance, feature visualization and robot learning.

**Strengths:**

- The overall paper is well written and motivated. It is clear what the goal of the paper is and why other approaches/datasets do not fulfill the requirements described in ArtVIP
- The dataset is crafted by experts following a specific assembly guideline, which should ensure that the objects are of high quality
- The dataset includes almost 1000 different assets which are articulated, as well as specific scenes and pixel-level annotations
- The evaluation of the dataset is thorough and includes different directions to showcase the advantages over other similar datasets
- Additional Imitation and Reinforcement Learning robot experiments solidify the usage of the dataset in some simple task settings
    - Results show that the Sim2Real gap is still given even with a more realistic dataset, but that it gets smaller

**Weaknesses:**

- Line 187-189: Any explicit source or statistic that confirms this?
- Visualized Feature Distribution: It is not clear from figure 5 on the right that ArtVIP object embeddings are actually that much closer to real world object embeddings. It looks more like they are still apart and ArtVIP is more closley related to OmniGibson. I think some other form of measurement for the feature distribution would be necessary.
- Claim (1) in line 430: Can you provide experiments using simulated data from other sources and show that the performance on the real robot is then actually worse? Otherwise it could be that ACT itself has some advantage for Sim2Real gap
- A comparison of time and money investment would be interesting compared to other datasets and also datasets which use learnable models to produce articulated objects
- It is also unclear how the higher polygon count affects the rendering speed. Additional comparisons of rollouts or training or robot models on the different asset datasets would help to understand the time difference for rendering.
- The dataset provides pixel-level affordance annotations, but there are no experiments showing the advantages of such an inclusion. Maybe you can train a vision-network to predict such affordances or use them for the robot learning models to better asses if an object is graspable or not. Further experiments would be helpful to verify the need for such annotations in this dataset.
- Minor Weaknesses
    - Figure 1 description is too short and uses an acronym which is not introduced before. I would also suggest to not use such a “teaser” figure, but rather put it below the abstract or even on top of page 2
    - Chapter 3 is an empty section
    - Figure 3 is never mentioned in the text
    - In Table 1 and 2 highlight important results with bold text or something similar

**Questions:**

- Can you provide additional robot experiments on other simulated datasets to verify the better performance of Sim2Real?
- How much slower is your approach in terms of rendering scenes given the higher polygon count?
- Why are you not evaluating the pixel-level annotations?
- It would be helpful if you can provide how much time and money in total the dataset needed.

---

> ### Author Response · Authors · 2025-11-27
>
> We thank the reviewer for the valuable feedback. Below, we provide detailed responses to the reviewer's primary comments and concerns.
> ### Q1&W3: Can you provide additional robot experiments on other simulated datasets to verify the better performance of Sim2Real?
> We added an experiment in Sec. 5.2: a cross‑dataset comparison against PartNet‑Mobility. We kept the original background unchanged. We selected five microwave ovens from PartNet‑Mobility and five from ArtVIP, and for the real‑world task we purchased an unseen microwave oven. The results are summarized in the table below. Higher‑fidelity ArtVIP assets yield stronger zero‑shot sim‑to‑real transfer and higher success under Real–Sim–Mixed settings, supporting the conclusion that higher‑quality assets reduce the sim‑to‑real gap and lead to higher sim‑to‑real success rates.
>
> | Method | Dataset    | Ours | PartNet-Mobility |
> |--------|------------|-----:|-----------------:|
> | ACT    | RO         | 56%   | 56%              |
> | ACT    | SO         | 41% (+9 pp)  | 32%              |
> | ACT    | RSM100+500 | 79% (+11 pp) | 68%              |
> | DP     | RO         | 62%   | 62%              |
> | DP     | SO         | 45% (+10 pp) | 35%              |
> | DP     | RSM100+500 | 83% (+13 pp) | 70%              |

---

> > ### Author Response · Authors · 2025-11-27
> >
> > ### Q2&W5: How much slower is your approach in terms of rendering scenes given the higher polygon count?
> > To study the effect of triangle count, we report comprehensive statistics in Tab.7 (Appendix) for ArtVIP, PartNet-Mobility, and BEHAVIOR-1K: the average triangle count, the average number of active joints, the average FPS with a single asset, and the average FPS in the kitchen scene. The kitchen scene is the most complex environment, containing 65 actuated joints. The evaluation results are as follows.
> >
> > | Item Category   | Single item (ArtVIP) | Single item (PartNet) | Single item (B1K) | Item in Kitchen (ArtVIP) | Item in Kitchen (PartNet) | Item in Kitchen (B1K) |
> > |-----------------|----------------:|---------------------------:|---------------------:|-----------------:|----------------------------:|----------------------:|
> > | Coffee Machine  | 91.97           | 91.95                      | 114.77               | 72.01            | 73.83                       | 48.33                 |
> > | Microwave       | 91.93           | 91.88                      | 87.51                | 69.38            | 72.95                       | 50.04                 |
> > | Oven            | 91.98           | 91.95                      | 109.97               | 72.66            | 74.64                       | 49.40                 |
> > | Dishwasher      | 91.94           | 91.95                      | 115.20               | 65.75            | 69.95                       | 49.73                 |
> > | Rice Cooker     | 91.96           | 91.96                      | 115.58               | 70.97            | 74.91                       | 48.29                 |
> > | Laptop          | 91.97           | 91.93                      | 112.00               | 74.59            | 74.89                       | 46.99                 |
> > | Washing Machine | 91.95           | 91.94                      | 107.17               | 70.97            | 70.74                       | 48.27                 |
> > | Toilet          | 91.95           | 91.95                      | 120.58               | 74.94            | 74.93                       | 47.18                 |
> > | Refrigerator    | 91.94           | 91.96                      | 99.55                | 60.78            | 73.89                       | 49.82                 |
> > | Table           | 91.96           | 91.96                      | 116.37               | 68.13            | 71.68                       | 47.92                 |
> > | Folding Chair   | 91.95           | 91.97                      | 125.93               | 74.50            | 74.92                       | 51.41                 |
> > | Scissors        | 91.96           | 91.96                      | 129.71               | 72.50            | 74.92                       | 52.45                 |
> > | Trash Can       | 91.94           | 91.93                      | 121.28               | 71.66            | 74.93                       | 52.24                 |
> >
> >
> > Based on the FPS results for ArtVIP and PartNet-Mobility, we conclude:
> > - For IL/RL, an FPS ideally above 60 is needed to match camera sampling frequency, but BEHAVIOR‑1K via OmniGibson does not meet this requirement.
> > - For a single object, under Isaac Sim's iterative optimizations, triangle counts up to approximately 100k and up to 20 active joints have negligible impact on FPS.
> > - In complex scenes, both triangle count and the number of active joints reduce FPS.
> >
> > Many factors affect simulation FPS, including polygon count, collision‑shape optimization, counts of static meshes and active rigid bodies, textures and materials, etc. Their impact is complicated; maintaining stable FPS therefore depends heavily on simulation engineering practices. Below are several practical optimizations that are not officially documented in Isaac Sim but have a large impact and implemented in ArtVIP:
> > 1. In articulated objects, minimize the number of active rigid‑body parts and use static meshes wherever parts never move. For example, cabinet tops and side panels are static, exclude them from kinematics calculation to improve simulation efficiency.
> > 2. Optimize collision geometry: use primitive shapes (e.g., cubes, cylinders) to replace complex collision meshes. This significantly improves collision‑detection efficiency and reduces computation.

---

> > > ### Author Response · Authors · 2025-11-27
> > >
> > > ### Q4&W4: It would be helpful if you can provide how much time and money in total the dataset needed.
> > > The total effort amounts to 2,977.3 hours (modeling plus physics tuning; per‑category breakdown in Tab. 4, Appendix). Unlike our dataset, other articulated‑object datasets[1,2,3] were not modeled from scratch under a unified standard, leading to uneven asset quality. We detail their shortcomings in next Weakness 1. Moreover, their purchase prices are not disclosed.
> > >
> > > ### W1: Line 187-189: Any explicit source or statistic that confirms "Existing datasets are largely sourced from pre-made models from public repositories"
> > > The following references confirm that existing datasets are largely sourced from pre-made models in public repositories.
> > > "The objects in the BEHAVIOR-1K dataset were sourced mainly through TurboSquid"[1]. In PartNet-Mobility, "all models are collected from 3D Warehouse"[2]. In RoboCasa, authors "gather object assets from two sources, the Objaverse[4] dataset and Luma.ai, an online text-to-3D service"[3]. RoboTwin "incorporates 153 objects from 27 categories in Objaverse [4], and 44 articulated object instances from 9 categories in SAPIEN PartNet-Mobility"[2].
> > > We summarize and visualize issues in several open-source datasets here, in order: [AdaManip](https://github.com/UejnYaM/digital_assets_comparisions/blob/main/AdaManip_assets_issues/details.md), [B1K](https://github.com/UejnYaM/digital_assets_comparisions/blob/main/b1k_assets_issues/b1k.md), and [PartNet-Mobility](https://github.com/UejnYaM/digital_assets_comparisions/blob/main/PartNet-Mobility_issues/PartNet-Mobility.md).
> > > In short, All AdaManip assets are completely unusable. B1K assets are low quality and have inconsistent local coordinate systems. PartNet‑Mobility has low mesh and texture quality.
> > >
> > > ### W2: Some other form of measurement for the feature distribution
> > > We have supplemented quantitative high-dimensional metrics in the same CLIP embedding space:
> > >
> > > | Metric | Real–OmniGibson | Real–ArtVIP |
> > > | --- | --- | --- |
> > > | Mean Cosine Distance (full pairwise) | 0.3895 | 0.3067 |
> > > | Mean Cosine Distance (nearest-neighbor matching) | 0.1742 | 0.1253 |
> > > | FID (unnormalized features) | 122.135 | 106.307 |
> > > | NNO (K=1, sample number=500) | 0.0 | 0.1 |
> > >
> > >
> > > We further conducted a significance test on the cosine distances using 500 bootstrapped samples: ArtVIP is significantly closer to real data than OmniGibson (**p = 0.008**).
> > > The results quantitatively confirm that ArtVIP is closer to real data than OmniGibson. We will include these metrics to complement the t-SNE visualization.
> > >
> > >
> > >
> > > References
> > > [1] https://behavior.stanford.edu/behavior_components/asset_sources.html
> > > [2] Xiang, Fanbo, et al. "Sapien: A simulated part-based interactive environment."
> > > [3] Nasiriany, Soroush, et al. "Robocasa: Large-scale simulation of everyday tasks for generalist robots."
> > > [4] Matt Deitke, et al."Objaverse: A universe of annotated 3d objects"
> > > [5] IIT‑AFF Dataset
> > > [6] UMD RGB‑D Part Affordance Dataset
> > > [7] Sawatzky, Srikantha, Gall. Weakly Supervised Affordance Detection (CVPR 2017)
> > > [8] Deng et al. 3D AffordanceNet (CVPR 2021)
> > > [9] Hussain et al. FPHA‑Afford (IEEE)
> > > [10] HOVA‑500K, a large‑scale affordance‑annotated dataset comprising 500,000 images and actions.” (Ning et al., 2025)
> > > [11] Chen et al. "MaskPrompt: Open‑Vocabulary Affordance Segmentation with Object Shape Mask Prompts" (AAAI 2025)
> > > [12] Li et al. "Learning Precise Affordances from Egocentric Videos for Robotic Manipulation" (ICCV 2025)
> > > [13] Kim et al. "ManipGPT: Is Affordance Segmentation by Large Vision Models Enough for Articulated Object Manipulation?"
> > > [14] "Let Me Show You: Learning by Retrieving from Egocentric Video for Robotic Manipulation"

---

### Official Review · Reviewer_NaQz · 2025-11-09

**Soundness:** 3
**Presentation:** 4
**Contribution:** 2
**Rating:** 4
**Confidence:** 4

**Summary:**

The paper proposes to create high-quality articulated object dataset using human modellers with an emphasis on quality than scaling. Qualitative and quantitative results show the scenes exhibit high photorealistic visual realism. The paper adopts a unified modelling and assembly guidelines given to human modellers and utilizes a top-down hierarchical mechanical modelling approach. Experiments are diverse and showcase various levels of improvements from visual realism compared to other sim provided assets, better intractability and well as imitation learning and reinforcement learning application for sim2real transfer.

**Strengths:**

In my opinion, below are the main strengths of the paper:

1. A high-quality 3D articulated object datasets combined with scene-level information i.e. kitchen etc. which exhibit greater photorealism and physical intractability.

2. Experiments are diverse and cover a breadth of tasks such as evaluating photorealism, interactability, reconstruction performance evaluation, feature distribution analysis as well as downstream application to imitation learning and RL.

3. The paper is nicely written, easy to follow and the figures/qualitative results nicely complement the text in the paper.

**Weaknesses:**

In my opinion, below are the main weakness in the paper:

1. While the qualitative results do show higher quality assets, it's unclear how well these fair when compared to other low-effort feed-forward approaches [1,2,3,4]. A comparison to these feed-forward baselines for the experiments outlines in the paper would justify the time spent in creating the higher quality assets where low-effort approaches sometime run at 1Hz for eight or less objects from a single RGB-D image [2].

2. While qualitative results are appreciated, it is not clear how well the method compare to existing democratized approaches [1,2,3,4] to articulated object generation interms of reconstruction performance i.e. with a chamfer distance metric to compare the geoemtric fidelity of the produced articulated assets.

3. For the imitation learning and RL results, it is unclear if the higher quality assets from the proposed approach led to higher success rate since these are RGB policies and it looks like significant effort was done in recreating table, robot as well as making some parts of the background similar to achieve sim2real transfer. Again, what if we replace the author's proposed assets with meshes from some of the feed-forward approaches, would it result in similar succes rate?

[Minor]

1. How much tuning of the parameters was carried out for interaction evaluation experiments. Can the same tuning effort be done for existing neural-network based articulated object generation approaches [1,2,3,4] and would it result in similar results?

2. Annotation time was not presented in the paper. Is this also significantly higher compared to other approaches that rely on distilled features etc? [5, 6]



[1] Liu et al. PARIS: Part-level Reconstruction and Motion Analysis for Articulated Objects, ICCV 2023
[2] Heppert et al, CARTO: Category and Joint Agnostic Reconstruction of ARTiculated Objects, CVPR 2023
[3] Lin et al. SplArt: Articulation Estimation and Part-Level Reconstruction with 3D Gaussian Splatting, ICCV 2025
[4] Mandi et al. Real2Code: Reconstruct Articulated Objects via Code Generation, ICLR 2025
[5] Yu et al. POGS: Persistent Object Gaussian Splat for Tracking Human and Robot Manipulation of Irregularly Shaped Objects, ICRA 2025
[6] Kerr et al. Robot See Robot Do Imitating Articulated Object Manipulation with Monocular 4D Reconstruction, COR 2024

**Questions:**

Please see questions in the weakness section, i look forward to seeing the author's response in the rebuttal.

---

> ### Author Response · Authors · 2025-11-27
>
> We thank the reviewer for the valuable feedback. Below, we provide detailed responses to the reviewer's primary comments and concerns.
> ### Q1: A comparison to other generative baselines
> #### Generative pipelines can't be integrated into simulation workflows:
> We reproduced SplArt [3] and generated assets for a two‑drawer cabinet and a side‑by‑side refrigerator. The resulting outputs are low‑quality and trigger runtime errors in Isaac Sim. Consequently, current generative baselines fail to produce simulation‑ready articulated assets for robotics. The process and output assets are available here. Image comparison is provided in Appendix Fig. 16.
> Process: [SplArt Experiment Records](https://github.com/UejnYaM/generative_pipelines_reproduce/blob/main/sqlgs/SplArt%20Experiment%20Records.md)
> Generated assets: [Cabinet (mp4)](https://github.com/UejnYaM/generative_pipelines_reproduce/blob/main/sqlgs/output/cabinet.mp4), [Fridge (mp4)](https://github.com/UejnYaM/generative_pipelines_reproduce/blob/main/sqlgs/output/fridge.mp4)
>
> From the above videos, the generated assets exhibit the issues of 1) self‑collisions, 2) severe mesh distortions and breakage, 3) incorrect joint limits, positions, and axes, 4) materials and colors that deviate markedly from reality, and 5) severe lack of interior details.
>
> #### Limitations of generative pipelines:
> Many baselines [1,3,4,7,8,9,10] are primarily trained and evaluated on synthetic data, where calibrated camera poses and controllable materials reduce domain gap. When performing inference on real images, they face:
> 1. Reconstruction accuracy degrades when camera extrinsics are estimated rather than known.
> 2. Since [1,3,4,7,8,9,10] are trained on PartNet‑Mobility, generalization is constrained by that dataset’s category coverage and articulation priors.
> 3. Flat surfaces and right‑angle structures often appear wavy or warped under monocular/low‑texture settings. (e.g., Figure 5 in Real2Code[4]).
> 4. Reconstruction outputs frequently lack physically consistent materials and textures. 3DGS methods encodes color via spherical harmonics (f_dc/f_rest) requiring custom shaders, so rendering pipeline simulators (Isaac Sim, Sapien, PyBullet, Mujoco, etc.) cannot render colors correctly without conversion/baking.
> 5. Current metrics do not capture mesh/triangle efficiency or collision performance relevant to simulation FPS.
>
>
> ### Q2: Quantitative evaluation of generated articulated assets
> We compare commonly used metrics for two real‑world objects (cabinet, fridge) against synthetic results reported in Tab.1 of ArtGS [7] to illustrate domain gap and failure cases in articulated reconstruction. Additionally, the generated cabinet and fridge contain ~100× more triangles than their ArtVIP counterparts, reducing the simulation frame rate from ~90 fps to ~70 fps.
>
> | Metric    | Real Cabinet | Real Fridge | Synthetic (ArtGS) Cabinet | Synthetic (ArtGS) Fridge |
> |-----------|--------:|-------:|:-----------|:----------|
> | CD-static |  13.71  |  11.04 | 7.31 | 0.52 |
> | CD-mobile |  11.50  |  13.45 | 1.02 | 0.27 |
> | CD-whole  |  12.35  |  12.78 | 5.21 | 0.7 |
> | Axis Ang  |  20.90  |  N/A | 0.01 |0.03 |
> | Axis Pos  |  0.094  |  N/A | - | 0.00 |
>
> Notes: CD denotes Chamfer Distance (↓ better). Axis Angular Error and Axis Positional Error are of the joint axis (↓ better). "N/A" indicates the axis estimate can not be produced (e.g., joint not detected).
>
>
> ### Q3: Did a similar background in simulation significantly contribute to the sim2real success rate?
> To verify that foreground asset quality rather than background similarity drives sim‑to‑real performance, we added an experiment in Sec. 5.2: a cross‑dataset comparison against PartNet‑Mobility. We kept the original background unchanged. We selected five microwave ovens from PartNet‑Mobility and five from ArtVIP, and for the real‑world task we purchased an unseen microwave oven. Higher‑fidelity ArtVIP assets yield stronger zero‑shot sim‑to‑real transfer and higher success under Real–Sim–Mixed settings, supporting the conclusion that higher‑quality assets reduce the sim‑to‑real gap and lead to higher success rates.
>
> | Method | Dataset    | Ours | PartNet-Mobility |
> |--------|------------|-----:|-----------------:|
> | ACT    | RO         | 56%   | 56%              |
> | ACT    | SO         | 41% (+9 pp)  | 32%              |
> | ACT    | RSM100+500 | 79% (+11 pp) | 68%              |
> | DP     | RO         | 62%   | 62%              |
> | DP     | SO         | 45% (+10 pp) | 35%              |
> | DP     | RSM100+500 | 83% (+13 pp) | 70%              |

---

> ### Author Response · Authors · 2025-11-27
>
> ### Q4: Can the same tuning effort be done for existing neural-network based articulated object generation approaches and would it result in similar results?
> Because assets produced by current generative pipelines exhibit systemic defects in geometric and physical consistency, they are typically not simulation‑ready and often raise runtime errors in Isaac Sim, as described in the answer to Q1. Even after comprehensive clean‑up and repair of generative assets, the time required to reach a “sim‑ready” standard is roughly comparable to manual modeling. These structural problems cannot be solved by physical parameter tuning alone. Taken together with the reproduction in Q1 and the quantitative results in Q2, generative assets struggle to reliably meet ArtVIP standards.
>
> ### Q5: Annotation time
> Annotation time is counted as part of modeling time in Tab. 4 (Appendix). We use an automatic script to add labels to 3D part models according to Tab. 5(Appendix), so the annotation time is too small to report separately.
>
>
> Reference:
> [1] Liu et al. PARIS: Part-level Reconstruction and Motion Analysis for Articulated Objects, ICCV 2023
> [2] Heppert et al, CARTO: Category and Joint Agnostic Reconstruction of ARTiculated Objects, CVPR 2023
> [3] Lin et al. SplArt: Articulation Estimation and Part-Level Reconstruction with 3D Gaussian Splatting, ICCV 2025
> [4] Mandi et al. Real2Code: Reconstruct Articulated Objects via Code Generation, ICLR 2025
> [5] Yu et al. POGS: Persistent Object Gaussian Splat for Tracking Human and Robot Manipulation of Irregularly Shaped Objects, ICRA 2025
> [6] Kerr et al. Robot See Robot Do Imitating Articulated Object Manipulation with Monocular 4D Reconstruction, COR 2024
> [7] Liu, Yu, et al. "Artgs: Building interactable replicas of complex articulated objects via gaussian splatting."
> [8] Guo, Junfu, et al. "Articulatedgs: Self-supervised digital twin modeling of articulated objects using 3d gaussian splatting."
> [9] Liu, Jiayi, et al. "Singapo: Single image controlled generation of articulated parts in objects."
> [10] Su, Jiayi, et al. "Artformer: Controllable generation of diverse 3d articulated objects."

---

### Official Review · Reviewer_oQH5 · 2025-11-10

**Soundness:** 3
**Presentation:** 3
**Contribution:** 2
**Rating:** 6
**Confidence:** 5

**Summary:**

The paper introduces ArtVIP, an open-source dataset of high-fidelity articulated digital-twin objects aimed at improving robot learning in simulation. Built by professional 3D modelers under unified standards, ArtVIP achieves visual realism through precise geometry and textures, and physical fidelity via tuned dynamic parameters. It further includes modular interaction behaviors and pixel-level affordance annotations. Quantitative evaluations and experiments in imitation and reinforcement learning confirm its strong sim-to-real transfer, making ArtVIP a valuable and reproducible resource for the robotics community.

**Strengths:**

1. The paper makes a valuable and practical contribution by releasing a high-quality articulated object dataset that combines visual realism, physical accuracy, and modular interactions. The modeling pipeline and embedded behaviors are clearly documented, ensuring long-term usefulness for the robotics community.

2. The dataset’s open-source release in USD format, along with conversion tools (URDF/MJCF) and comprehensive production guidelines, greatly improves accessibility, reproducibility, and integration into diverse simulation workflows.

3. The imitation learning and reinforcement learning experiments convincingly demonstrate strong sim-to-real generalization, underscoring the dataset’s effectiveness for both visual and physical sim-to-real transfer.

4. The figures are clear and motivating, effectively conveying the dataset’s realism and interactivity.

**Weaknesses:**

1. The primary contribution lies in dataset engineering rather than methodological novelty. While the dataset’s quality is commendable, its scalability is constrained by manual modeling and tuning, which may limit extensibility. With the rise of generative pipelines such as RoboTwin[1] and Genesis[2], ArtVIP’s labor-intensive approach appears less sustainable for expansion.

2. The claimed physical fidelity mainly covers joint parameters such as damping, friction, and magnetic closure, but overlooks more complex mechanical dependencies within articulated systems. Works like AdaManip [3] and DoorGym [4] emphasize such mechanisms. For instance, rotating a doorknob before unlatching or twisting a pressure-cooker lid before lifting are emphasized in these works. These multi-stage kinematic couplings represent realistic physical constraints that ArtVIP does not yet model.

3. Although the paper includes imitation learning and RL experiments, the evaluation scope remains narrow, focusing primarily on sim-to-real ratios within ArtVIP. Direct benchmarks against other datasets (e.g., RoboTwin, AdaManip) under identical task setups would more convincingly demonstrate ArtVIP’s advantages and generality.

[1] Chen, Tianxing, et al. "Robotwin 2.0: A scalable data generator and benchmark with strong domain randomization for robust bimanual robotic manipulation." arXiv preprint arXiv:2506.18088 (2025).

[2] Zhou, Xian, et al. "Genesis: A Generative and Universal Physics Engine for Robotics and Beyond." arXiv preprint arXiv:2401.01454 (2024).

[3] Wang, Yuanfei, et al. "Adamanip: Adaptive articulated object manipulation environments and policy learning." arXiv preprint arXiv:2502.11124 (2025).

[4] Urakami, Yusuke, et al. "Doorgym: A scalable door opening environment and baseline agent." arXiv preprint arXiv:1908.01887 (2019).

**Questions:**

1. Could the authors elaborate on how ArtVIP might scale to larger datasets or environments in the future? Given the heavy reliance on manual modeling, would integrating generative pipelines be a feasible direction for expansion?

2. How does ArtVIP compare with prior works that model realistic physical mechanisms rather than focusing mainly on joint parameters?

3. Would it be possible to include cross-dataset or benchmark comparisons (e.g., with RoboTwin or AdaManip) under identical task settings to better contextualize ArtVIP’s contribution?

---

> ### Author Response · Authors · 2025-11-27
>
> We thank the reviewer for the valuable feedback. Below, we provide detailed responses to the reviewer's primary comments and concerns.
>
> ### Q1&W1: How to scale up ArtVIP and why not integrating generative pipelines?
> During the development of ArtVIP, we summarized methods that streamline workflow and reduce costs, and applied them in recent production runs, yielding promising results.
> We have not integrated generative pipelines because the generated assets are low‑quality and often cause simulation errors at the current stage.
>
> #### 1.Scaling via principled optimizations:
> During the development of ArtVIP, after producing the first ~500 assets, the average per‑asset time falls by ~20–30%, and after ~1,000 assets it declines by ~50–60%, showing efficient dataset scaling. These gains are driven by:
> - Built a reusable library of 3D parts and textures. As the library expands, modeling time continuously decreases.
> - Modularized interaction components and auto‑generated scripts for physical parameter tuning, reducing tuning time by ~90%.
>
> #### 2.Generative pipelines can't be integrated into simulation workflows:
> We reproduced SplArt [7] and generated assets for a two‑drawer cabinet and a side‑by‑side refrigerator. The resulting outputs are low‑quality and trigger runtime errors in Isaac Sim. Consequently, current generative baselines fail to produce simulation‑ready articulated assets for robotics. The process and output assets are available here. Image comparison is provided in Appendix Fig. 16.
> Process: [SplArt Experiment Records](https://github.com/UejnYaM/generative_pipelines_reproduce/blob/main/sqlgs/SplArt%20Experiment%20Records.md)
> Generated assets: [Cabinet (mp4)](https://github.com/UejnYaM/generative_pipelines_reproduce/blob/main/sqlgs/output/cabinet.mp4), [Fridge (mp4)](https://github.com/UejnYaM/generative_pipelines_reproduce/blob/main/sqlgs/output/fridge.mp4)
>
> From the above videos, the generated assets exhibit the issues of 1) self‑collisions, 2) severe mesh distortions and breakage, 3) incorrect joint limits, positions, and axes, 4) materials and colors that deviate markedly from reality, and 5) severe lack of interior details.
>
> We compare commonly used metrics for two real‑world objects (cabinet, fridge) against synthetic results reported in Tab.1 of ArtGS [9] to illustrate domain gap and failure cases in articulated reconstruction. Additionally, the generated cabinet and fridge contain ~100× more triangles than their ArtVIP counterparts, reducing the simulation frame rate from ~90 fps to ~70 fps.
>
> | Metric    | Real Cabinet | Real Fridge | Synthetic (ArtGS) Cabinet | Synthetic (ArtGS) Fridge |
> |-----------|--------:|-------:|:-----------|:----------|
> | CD-static |  13.71  |  11.04 | 7.31 | 0.52 |
> | CD-mobile |  11.50  |  13.45 | 1.02 | 0.27 |
> | CD-whole  |  12.35  |  12.78 | 5.21 | 0.7 |
> | Axis Ang  |  20.90  |  N/A | 0.01 |0.03 |
> | Axis Pos  |  0.094  |  N/A | - | 0.00 |
>
> Notes: CD denotes Chamfer Distance (↓ better). Axis Angular Error and Axis Positional Error are of the joint axis (↓ better). "N/A" indicates the axis estimate can not be produced (e.g., joint not detected).
>
> #### 3.Compare with generative pipelines such as RoboTwin[1] and Genesis[2]:
> RoboTwin [1] and Genesis [2] are simulation platforms that do not generate usable articulated objects, which is the core contribution of ArtVIP.
>
> RoboTwin’s generative pipelines produce only rigid bodies. RoboTwin [1] states that all articulated objects are sourced from PartNet‑Mobility [5], while static rigid objects come from Objaverse [6] and the 3D reconstruction Rodin platform (https://hyper3d.ai/). In other words, RoboTwin does not use generative pipelines to create articulated objects.
>
> As for Genesis [2], although the official website showcases animations of articulated objects, neither assets nor code are open‑sourced, and no reproduction procedure for its generative pipeline is publicly documented. Consequently, we cannot perform a reproducible comparison. The animations suggest axis misalignment for the car wheels, power drill, and blender, indicating low mesh precision. The highlight on the Poké Ball rotates with the ball, implying incorrect material/lighting and unrealistic rendering. Therefore, Genesis’s assets do not meet ArtVIP’s quality requirements.

---

> > ### Author Response · Authors · 2025-11-27
> >
> > #### 4.Limitations of generative pipelines:
> > Many baselines [7~12] are primarily trained and evaluated on synthetic data, where calibrated camera poses and controllable materials reduce domain gap. When performing inference on real images, they face:
> > 1. Reconstruction accuracy degrades when camera extrinsics are estimated rather than known.
> > 2. Since [7~12] are trained on PartNet‑Mobility, generalization is constrained by that dataset’s category coverage and articulation priors.
> > 3. Flat surfaces and right‑angle structures often appear wavy or warped under monocular/low‑texture settings. (e.g., Figure 5 in Real2Code[8]).
> > 4. Reconstruction outputs frequently lack physically consistent materials and textures. 3DGS methods encodes color via spherical harmonics (f_dc/f_rest) requiring custom shaders, so rendering pipeline simulators (Isaac Sim, Sapien, PyBullet, Mujoco, etc.) cannot render colors correctly without conversion/baking.
> > 5. Current metrics do not capture mesh/triangle efficiency or collision performance relevant to simulation FPS.

---

> ### Author Response · Authors · 2025-11-27
>
> ### Q2&W2: Compare with prior works that model realistic physical mechanisms
> ArtVIP encompasses **all realistic physical mechanisms** implemented in AdaManip and DoorGym, delivers higher joint motion fidelity, and is portable to other benchmarks (e.g., any Isaac Sim–based setup).
>
> In our initial submission, we summarized common realistic physical mechanisms for articulated objects into five cases. Fig. 3 presents simulation frames illustrating their dynamics, and Sec. 3.4 explains the implementation principles. We also include demonstration videos of these mechanisms in the uploaded materials folder `3.modular_interactive_objects`.
>
> In terms of realism, ArtVIP not only implements realistic physical mechanisms but also dynamically adjusts joint stiffness, damping, and drive in real time during simulation (Appendix Sec. E details the fine‑tuning formulas), producing more faithful dynamics. For example, when the doorknob is rotated, the door exhibits slight damping that decreases as it opens, consistent with real‑world behavior. By contrast, AdaManip and DoorGym use fixed physical parameters (e.g., stiffness, damping).
>
> In terms of portability, AdaManip and DoorGym implement these mechanisms within algorithm code. When assets are used in other benchmarks decoupled from the original algorithms, reusing the mechanisms increases development complexity and time. ArtVIP embeds the mechanisms into the assets via Modular Interaction, enabling users to reproduce them by simply importing the assets, no additional code required.
>
> We compare ArtVIP’s mechanisms with those in AdaManip and DoorGym.
> | Task | AdaManip/DoorGym mechanisms | ArtVIP mechanisms | Corresponding ArtVIP technique | Comparison |
> |--|--|--|--|--|
> | open_bottle, open_pen | Simulate cap–body separation via a prismatic joint | Model cap rotation using realistic collision | Physical fidelity | ArtVIP is more realistic |
> | open_coffeemachine | Use a revolute joint to simulate portafilter rotation | Use a revolute joint to simulate portafilter rotation | Within‑asset effects | Same mechanism |
> | open_door, open_window, open_safe| Door rotation controlled by the knob joint position | Door rotation controlled by the knob joint position | Within‑asset effects | Same mechanism |
> | open_lamp | Only supports switching on | Use the same switch to toggle both on/off | Cross‑asset effects | ArtVIP provides bidirectional switching, whereas AdaManip supports only "on" |
> | open_microwave | No interior light | Button press opens door and turns light on | Within‑asset effects | ArtVIP couples door and light, whereas AdaManip lacks interior light |
> | open_pressurecooker | Simulate lid–body separation via a prismatic joint | Simulate lid–body separation via a prismatic joint | Within‑asset effects | Same mechanism |

---

> ### Author Response · Authors · 2025-11-27
>
> ### Q3&W3: Cross-dataset or benchmark comparisons (e.g., with RoboTwin or AdaManip) under identical task settings
> We added an experiment in Sec. 5.2: a cross‑dataset comparison against PartNet‑Mobility, which is the source of articulated objects used in RoboTwin. We kept the original background unchanged. We selected five microwave ovens from PartNet‑Mobility and five from ArtVIP, and for the real‑world task we purchased an unseen microwave oven. The results are summarized in the table below. We found that higher‑fidelity ArtVIP assets yield stronger zero‑shot sim‑to‑real transfer and higher success under Real–Sim–Mixed settings, supporting the conclusion that higher‑quality assets reduce the sim‑to‑real gap and lead to higher success rates.
>
> | Method | Dataset    | Ours | PartNet-Mobility |
> |--------|------------|-----:|-----------------:|
> | ACT    | RO         | 56%   | 56%              |
> | ACT    | SO         | 41% (+9 pp)  | 32%              |
> | ACT    | RSM100+500 | 79% (+11 pp) | 68%              |
> | DP     | RO         | 62%   | 62%              |
> | DP     | SO         | 45% (+10 pp) | 35%              |
> | DP     | RSM100+500 | 83% (+13 pp) | 70%              |
>
> We attempted to reproduce the AdaManip assets. Due to joint coordinate system issues, all assets are completely unusable. A visualization of issues in the AdaManip URDF files is available [here](https://github.com/UejnYaM/digital_assets_comparisions/blob/main/AdaManip_assets_issues/details.md).
> RoboTwin "incorporates 153 objects from 27 categories in Objaverse [6], and 44 articulated object instances from 9 categories in SAPIEN PartNet‑Mobility" [5]. Therefore, we directly compare assets from PartNet‑Mobility and ArtVIP.
>
>
> Reference:
> [1] Chen, Tianxing, et al. "Robotwin 2.0: A scalable data generator and benchmark with strong domain randomization for robust bimanual robotic manipulation." arXiv preprint arXiv:2506.18088 (2025).
> [2] Zhou, Xian, et al. "Genesis: A Generative and Universal Physics Engine for Robotics and Beyond." arXiv preprint arXiv:2401.01454 (2024).
> [3] Wang, Yuanfei, et al. "Adamanip: Adaptive articulated object manipulation environments and policy learning." arXiv preprint arXiv:2502.11124 (2025).
> [4] Urakami, Yusuke, et al. "Doorgym: A scalable door opening environment and baseline agent." arXiv preprint arXiv:1908.01887 (2019).
> [5] Xiang, Fanbo, et al. "Sapien: A simulated part-based interactive environment."
> [6] Matt Deitke, et al."Objaverse: A universe of annotated 3d objects"
> [7] Lin et al. SplArt: Articulation Estimation and Part-Level Reconstruction with 3D Gaussian Splatting, ICCV 2025
> [8] Mandi et al. Real2Code: Reconstruct Articulated Objects via Code Generation, ICLR 2025
> [9] Liu, Yu, et al. "Artgs: Building interactable replicas of complex articulated objects via gaussian splatting."
> [10] Guo, Junfu, et al. "Articulatedgs: Self-supervised digital twin modeling of articulated objects using 3d gaussian splatting."
> [11] Liu, Jiayi, et al. "Singapo: Single image controlled generation of articulated parts in objects."
> [12] Su, Jiayi, et al. "Artformer: Controllable generation of diverse 3d articulated objects."

---

> > ### Comment · Reviewer_oQH5 · 2025-11-28
> > **Reply to Authors**
> >
> > Thanks for providing the additional experimental results and discussions, which improve the significance and soundness of the paper. I suggest the authors include these clarifications and results in the revised version of the paper, and I support acceptance of the paper now.

---

### Author Response · Authors · 2025-12-03
**General Response to Reviewers**

We thank the Reviewers, Area Chairs, and Program Chairs for their time and thoughtful feedback. We have carefully considered all comments and provided detailed responses under each Reviewer’s section.

We are encouraged by the Reviewers’ recognition of ArtVIP as a high‑quality, simulation‑ready solution tailored for complex robotic manipulation, highlighted across three key dimensions:
- Innovation:
  - Fidelity: "a valuable and practical contribution"(`reviewer oQH5`); "exhibit greater photorealism and physical interactivity"(`reviewer NaQz`); "ensure that the objects are of high quality"(`reviewer 1Zfo`)
  - Interactivity: "encoding interactive semantics directly in USD assets is novel and practically useful"(`reviewer YbiS`); "embedding customizable, reusable behaviors directly within assets is novel and highly beneficial"(`reviewer CLY8`)
- Evaluation: "experiments convincingly demonstrate strong sim-to-real generalization"(`reviewer oQH5`), "experiments are diverse and cover a breadth of tasks"(`reviewer NaQz`); "comprehensive evaluations demonstrate practical utility"(`reviewer jg1J`); "evaluation is thorough"(`reviewer 1Zfo`); "the analysis is comprehensive"(`reviewer YbiS`)
- Presentation: "well written and motivated"(`reviewer 1Zfo`); "figures/qualitative results nicely complement the text"(`reviewer NaQz`); "figures are clear and motivating"(`reviewer oQH5`); "The paper is very well written"(`reviewer YbiS`)

Reviewers emphasized the high quality of ArtVIP assets and the innovative interactivity, and recognized our IL/RL evaluation. The main concerns centered on (1) comparisons using simulated data from other datasets; and (2) why ArtVIP does not integrate generative pipelines for scaling.
To address these points, we added experiments and submitted a revised paper. Summary:
- Section 5.2 (main results)

  Addresses questions from `reviewer oQH5`, `reviewer NaQz`, `reviewer 1Zfo`, `reviewer YbiS`. We added cross‑dataset sim‑to‑real results showing that high‑quality ArtVIP assets effectively reduce the sim‑to‑real gap. In zero‑shot sim‑to‑real experiments, ArtVIP outperforms PartNet‑Mobility by 28% in success rate.
- Appendix K (generative pipelines comparison)

  Addresses questions from `reviewer oQH5`, `reviewer NaQz`, `reviewer YbiS`, `reviewer CLY8`. We reproduced a SOTA generative method and found that the generated assets are of low quality and not simulation-ready, frequently causing simulation errors and resulting in a 0% task success rate.
- Appendix G (statistics)

  Addresses questions from `reviewer 1Zfo`, `reviewer CLY8`. We added statistics and analysis of triangle counts and simulation FPS. Although ArtVIP employs a higher triangle count to enhance asset quality, our optimizations ensure that all complex scenes maintain a simulation FPS of at least 60.
- Section 5.3 (experiment explanation)

  Addresses questions from `reviewer jg1J`. We added the Pearson correlation formula and clarified its computation.
- Minor corrections & terminology

  We now reference Figure 3 in Section 3.3 and, following `reviewer 1Zfo`'s suggestion, adjusted the placement of the teaser figure.

In short, ArtVIP addresses the shortage of high‑quality simulation assets with professionally crafted digital twins. Its unified standards for realism, fidelity, and embedded behaviors enable complex manipulation and substantially bridge the sim‑to‑real gap.

We sincerely thank the reviewers for their valuable feedback and suggestions, which helped us strengthen the paper.


Best regards,

Authors of Submission 16279

---

### Meta-Review · Area_Chair_t5gB · 2026-01-06

**Summary:**

There is an important gap in simulation‑ready articulated assets for robotics. This paper ArtVIP introduces a large‑scale, professionally modeled dataset of nearly 1000 articulated digital‑twin objects designed to improve sim‑to‑real transfer in robot learning by combining high visual realism, precise physical fidelity, and embedded modular interaction behaviors. Reviewers appreciated the dataset’s high asset quality, photorealistic textures, accurate joint dynamics, and novel integration of interaction semantics directly into USD assets, which enables zero‑code reuse across tasks. They highlighted the comprehensive evaluation, including reconstruction fidelity, feature‑distribution analysis, optical motion‑capture validation, and strong IL/RL sim‑to‑real results.

The paper was thoroughly reviewed by six reviewers! The following main concerns were raised:
- Lack of comparison to generative pipelines / feed‑forward articulated reconstruction methods
- Scalability concerns due to manual modeling cost (3000 hours of manual effort for the 992 assets)
- Insufficient cross‑dataset comparisons (e.g., PartNet‑Mobility, RoboTwin, AdaManip)
- Unclear whether background similarity drives sim‑to‑real, unclear contribution of each component (visual realism, physical fidelity, modular interaction)
- cost / time breakdown
- Missing evaluation of pixel‑level affordances
- Scalability concerns due to manual modeling cost
- Dataset size modest (992 objects)

**Reviewer Concerns:**

Addressed concerns:
- Lack of comparison to generative pipelines / feed‑forward articulated reconstruction methods: Reproduced SplArt; showed severe mesh distortions, collisions, incorrect joints, unrealistic materials; generative assets not simulation‑ready; FPS drops; metrics show large domain gap; thus cannot be integrated.
- Insufficient cross‑dataset comparisons (e.g., PartNet‑Mobility, RoboTwin, AdaManip)"
- Unclear whether background similarity drives sim‑to‑real, unclear contribution of each component (visual realism, physical fidelity, modular interaction)
- cost / time breakdown:

Partially resolved / unresolved concerns:

- Missing evaluation of pixel‑level affordances: Not directly evaluated; authors note annotations exist but experiments focus on IL/RL.
- Scalability concerns due to manual modeling cost: Authors point out that modeling time drops as more assets are added, due to part libraries etc.
- Dataset size modest (992 objects): authors point out that this is large by today's standards.

**Reviewer Scores:**

oQH5: 6 -> 6/8
NaQz: 4 -> 6
1ZFO - 6 -> 6
YbiS: 8 -> 8
jg1J: 2 -> 4
CLY8: 6 -> 6

---

### Decision · Program_Chairs · 2026-01-26

Accept (Poster)